# Dynamic WT1 expression during gastrulation specifies peritoneal smooth muscle fate independently of mesothelial fate

Suad H. Alsukari[1,*,‡], Huei Teng Ng[1,‡], Lilly Lang[1], Sharna Lunn[2], Shanthi Beglinger[1], Lauren Carr[1], Michael Boyes[1], David A. Turner[2] and Bettina Wilm[1,§]

## ABSTRACT

The Wilms' tumor protein (WT1) was previously linked to the mesothelial and vascular smooth muscle cell (vSMC) lineage in the mouse intestine, with evidence suggesting that intestinal vSMCs arise from the mesothelium. Here, we report that WT1 is already expressed, unexpectedly, during mouse gastrulation, in cells that specify a population of SMC precursor cells in the lateral plate mesoderm. Tamoxifen-induced genetic lineage tracing of *Wt1*-expressing cells revealed that vSMC and visceral smooth muscle cells (visSMCs) of the foetal mid-gut, but not mesothelial cells, were labelled after tamoxifen administration at embryonic day (E) 7.5 or E8.5. Analysis of single-cell RNA sequencing datasets of gastrulation-stage mouse embryos and confocal microscopy demonstrated *Wt1* expression in epiblast, primitive streak, emerging mesoderm and, from E7.5 onwards, in the lateral plate mesoderm. Co-expression of signature smooth muscle markers in *Wt1*-expressing cells in gastrulation-stage embryos revealed that vSMC and visSMC fate is specified independently of visceral mesothelium formation. Furthermore, tamoxifen-induced *Wt1* knock out at E7.5 affected vascularisation in the E12.5 intestine. Taken together, our study provides previously unknown insights into the developmental lineage of smooth muscle specified by WT1 expression during gastrulation.

KEY WORDS: Mouse gastrulation, Wilms' tumor protein 1, Mesodermal cell fate, Smooth muscle, Mesothelium

## INTRODUCTION

The embryonic origin of vascular smooth muscle cells (vSMCs) is relatively diverse as different regions of the vasculature arise from different progenitors, including neural crest cells, lateral plate mesoderm (LPM) and somatic mesoderm (Donadon and Santoro, 2021). Differentiation of vSMCs and tissue formation in vertebrates are driven by a series of canonical cell signalling events and mechanical

[1]Department of Women's and Children's Health, Institute of Life Course and Medical Sciences, Faculty of Health and Life Sciences, University of Liverpool, Liverpool L69 3BX, UK. [2]Department of Musculoskeletal and Ageing Science, Institute of Life Course and Medical Sciences, Faculty of Health and Life Sciences, University of Liverpool, Liverpool L7 8TX, UK.
*Present address: Department of Biology, Faculty of Science, Al-Baha University, P.O. Box 1988, Al-Baha 65799, Saudi Arabia.
‡These authors contributed equally to this work

§Author for correspondence (bwilm@liverpool.ac.uk)

H.T.N., 0000-0002-8785-5356; S.L., 0000-0002-6497-6649; D.A.T., 0000-0002-3447-7662; B.W., 0000-0002-9245-993X

forces that regulate progenitor cell recruitment, differentiation and maturation (del Monte et al., 2011; Qi et al., 2021; Stratman et al., 2020). Platelet-derived growth factor receptor beta (PDGFRB) signalling is a key regulatory pathway in vSMC myogenesis since disruption of PDGFRB signalling reduces vSMC proliferation and recruitment (Hellström et al., 1999; Stratman et al., 2020). Specification of vSMCs is marked by expression of smooth muscle cell markers, such as transgelin (TAGLN), alpha-smooth muscle actin (ACTA2), smooth muscle myosin heavy chain (MYH11), desmin and others (Assinder et al., 2009; Muhl et al., 2022; Smyth et al., 2018). However, the time point at which these genes start to be expressed in embryonic development, and their roles during the early stages of vSMC specification and differentiation, remain unclear.

Wilms' tumor protein 1 (WT1) is a transcription factor involved in the development of several tissues and organs in normal embryogenesis. The earliest WT1 expression was identified in the intermediate mesoderm (IM) and LPM in embryonic day (E) 9.5 mouse embryos (Armstrong et al., 1993). Its most prominent roles are in the development of the kidneys, arising from the IM, and of the heart, where the WT1-expressing epicardium contributes to coronary vessel formation (Kreidberg et al., 1993; Rackley et al., 1993). The epicardium is the mesothelial layer of the heart, which is continuous with the pericardial mesothelium of the pericardial cavity. WT1 is also expressed in the pleural and peritoneal mesothelium, lining the cavities and covering the organs housed within, from about E9.5 onwards and throughout adulthood (Armstrong et al., 1993; Moore et al., 1999, 1998; Que et al., 2008; Wilm et al., 2005, 2021).

Previously, studies showed that the epicardium can give rise to vSMCs of the developing coronary vasculature (Guadix et al., 2006; Zhou et al., 2008). Genetic lineage-tracing experiments using a transgenic mouse line expressing Cre recombinase under control of regulatory elements of the human *WT1* gene [*Tg(WT1-cre)AG11Dbdr*; abbreviated as *WT1-Cre*] in combination with the *Rosa26^{lacZ/lacZ}* reporter mouse revealed that a population of WT1-expressing serosa and pleural mesothelium cells give rise to vSMCs in the peritoneal and lung cavities (Que et al., 2008; Wilm et al., 2005). These results suggested that the mesothelium and vasculature of the intestine and lungs may have a developmental relationship similar to that of the epicardium and the coronary vasculature. Similar studies using *mWt1/IRES/GFP-Cre* (*Wt1cre*) transgenic mice crossed with the *Rosa26R^{EYFP}* reporter mouse also found a contribution of WT1-expressing mesothelial cells (MCs) to the vasculature of the developing intestine (Carmona et al., 2013).

In order to determine when MCs become committed to give rise to intestinal vascular smooth muscle, we previously conducted pulse-chase studies of postnatal and adult mice (Wilm et al., 2021) using a tamoxifen (Tam)-inducible, temporally controlled and *Wt1*-driven Cre system [*Wt1^{tm2(cre/ERT2)Wtp}*, *Wt1^{CreERT2}*; Zhou et al., 2008]. Our results showed that WT1-expressing peritoneal cells maintained the mesothelium over the intestine and peritoneal wall

(and contributed to visceral fat; Chau et al., 2011), but failed to give rise to intestinal vasculature components. However, WT1-expressing epicardium was observed to contribute towards formation of coronary vessels postnatally and in adult mice (Wilm et al., 2021). These findings suggested that the capacity of WT1-expressing MCs within the peritoneal cavity to differentiate into vSMCs must be restricted to developmental stages.

We therefore propose that there is an embryonic contribution of WT1-expressing cells towards the vSMC fate in the peritoneum and intestine, of which the temporo-spatial resolution and molecular mechanisms have remained unresolved so far. Here, to address this, we used the Tam-inducible $Wt1^{CreERT2}$ mouse line in combination with Rosa-based reporter mice for lineage-tracing analysis to determine the contribution of WT1-expressing cells towards intestinal vasculature in the mouse embryo. Our results indicate that, during embryonic development, two distinct populations of WT1-expressing cells emerge in the peritoneum, one that defines MCs, and one that defines SMC fate. Lineage-traced cells were found not only as vSMCs but also as visceral SMCs (visSMCs) in the intestinal wall, when Tam had been given at stages around E7.5 to E8.5, while predominantly MCs were labelled when Tam had been given from approximately E9.5 onwards. Importantly, both vSMCs and visSMCs, as well as the general parietal peritoneal wall tissue including the mesothelium, is derived from the LPM.

We provide, for the first time, a detailed analysis of WT1 transcript and protein expression in E6.5-E8.5 mouse embryos. In silico analysis of single-cell RNA sequencing (scRNAseq) data sets revealed that $Wt1$ is co-expressed with the emerging mesoderm marker brachyury ($Bra$; $T$) during gastrulation, and later with genes representing an SMC signature in E6.5-E8.5 mouse embryos. We then describe expression of WT1 in the E7.5 and E8.5 mouse embryo in the primitive streak and the emerging neuro-mesodermal progenitor cells as well as LPM using cutting-edge confocal microscopy techniques. WT1-derived cells within the mesentery of the developing intestine aligned with the developing endothelial vascular network and co-expressed PDGFRB, while inactivation of WT1 affected vascularisation of the developing intestine. Together, these data reveal a previously unappreciated developmental differentiation pathway of visceral and vascular smooth muscle cells influenced by WT1 expression that is independent of mesothelium formation.

## RESULTS
### Contribution of *Wt1*-expressing cells to SMCs in the peritoneal cavity

To determine when during embryogenesis Wt1-expressing cells within the peritoneal cavity became fated to develop into vSMCs, we combined the $Wt1^{CreERT2/+}$ lineage driver with either the $Rosa26R$ ($lacZ$) or the $Rosa26^{mTmG}$ (GFP-fluorescent) lineage reporter in mice, gave Tam once at different embryonic stages (pulse), and performed endpoint analysis at stages around E18.5 (chase; see Fig. 1A, Fig. S1A, Table S1). We obtained similar results with both reporter systems: Tam administration at E9.5 or up to E13.5, followed by endpoint whole-mount analysis, revealed almost exclusive labelling of cells covering the surface of the mesentery and intestine at mid-intestinal level (Fig. 1B-E, Fig. S1B-F, Table S1). Based on their position and shape, we concluded that these were MCs. When Tam was given to pregnant dams at E7.5 or E8.5 followed by endpoint analysis, we observed instead labelled cells surrounding the developing vessels of the mesentery, and in what appeared to be the circular visceral smooth muscle of the intestine at mid-intestinal level, but barely in the mesothelium (Fig. 1F-J, Fig. S1G-K). Littermates carrying no $Wt1^{CreERT2}$ allele showed no labelled cells after Tam administration (Fig. S1L,M).

We also analysed the stomach with attached spleen and the colon of foetuses under this experimental design, and observed that Tam dosing at E7.5 resulted in labelling of cells with circular visceral smooth muscle appearance but not surface-covering MCs. By contrast, foetuses after Tam dosing at E8.5 showed a mixed population of labelled cells in the stomach, mostly consisting of what appeared to be circular visceral smooth muscle; in the colon, there were only labelled cells with circular visceral smooth muscle appearance. In foetuses that had received Tam at E9.5 up to E14.5, cell labelling occurred predominantly in surface-covering MCs (Fig. S2, Table S1). However, after Tam dosing at E9.5 and up to E11.5 we observed in a few foetuses occasional $lacZ$-expressing cells that appeared to be circular visceral smooth muscle, in the stomach and colon.

Interestingly, the spleen, which is also an LPM-derived organ, was not labelled when Tam was administered at E7.5 (Fig. S2A), but was strongly labelled in foetuses with Tam dosing from E8.5 onwards (Fig. S2B-F). This indicates that WT1 was not yet expressed at E7.5 in the cells that later will become splenic precursors forming the putative splenic mesenchyme (Hecksher-Sørensen et al., 2004).

Immunofluorescence (IF) staining on sections through intestine and mesentery of $Wt1^{CreERT2/+}$;$Rosa26^{mTmG/+}$ foetuses (Fig. 2J) confirmed that after inducing Tam-based lineage tracing at E8.5, visSMCs and vSMCs co-expressed ACTA2 and GFP and were found close to the CD31 (PECAM1)-positive vascular endothelial cells, while hardly any MCs (pan-cytokeratin, WT1) were GFP positive (Fig. 2A-E′). By contrast, after Tam dosing at E11.5, MCs were GFP-labelled but no visSMCs or vSMCs (Fig. 2F-I′). These results suggest that, during embryonic development, two distinct populations of WT1-expressing cells emerge: one that defines MCs from approximately E9.5 onwards, and one that defines visceral and vascular SMC fate predominantly at stages around E7.5 to E8.5.

An additional observation in the mice after Tam dosing at E7.5 and E8.5 was that labelled visSMCs and vSMCs could only be found in a short fragment of the mid-intestinal region including mesentery (Fig. S1G). This suggests a short pulse-like or transient period of WT1 expression during which Tam exposure allowed the recombination and labelling of future visSMCs and vSMCs cells in the midgut region. However, the labelled visSMCs in the stomach and colon additionally indicate contribution along fore- and hindgut regions not limited to initiation at E7.5-E8.5.

### *Wt1* transcript is expressed from E6.5 in mouse embryos and co-expressed with markers for smooth muscle lineage

Next, we analysed expression of $Wt1$ transcripts during mouse gastrulation stages E6.5 to E8.5 using a published scRNAseq data set (Pijuan-Sala et al., 2019). We found that $Wt1$ was expressed in mouse embryos throughout this developmental period in 0.65-1.66% of total embryonic cells (Fig. S3A). Mean expression for $Wt1$ was between 0.955 and 1.13 with cells considered as expressing $Wt1$ if levels were above 0 [expression levels were log-normalised (count per million)] (Fig. S3B). $Wt1$ expression was highly dynamic, with the majority of $Wt1$-expressing cells between E6.5 and E7.25 identified as epiblast, and from E7.5 onwards shifting towards a majority in cells described as 'mesenchyme' (from E8.25-E8.5; Fig. S3C). Since visSMCs and vSMCs have their origin in the LPM but the dataset annotation lacked a clear indication which of the mesoderm tissues were LPM, we characterised in a candidate approach somatic, paraxial, nascent, mixed and intermediate mesoderm together with primitive streak, notochord, epiblast and mesenchyme for classical LPM and other mesodermal markers, confirming that 'mesenchyme' was representing the LPM population (Fig. S3E,F).

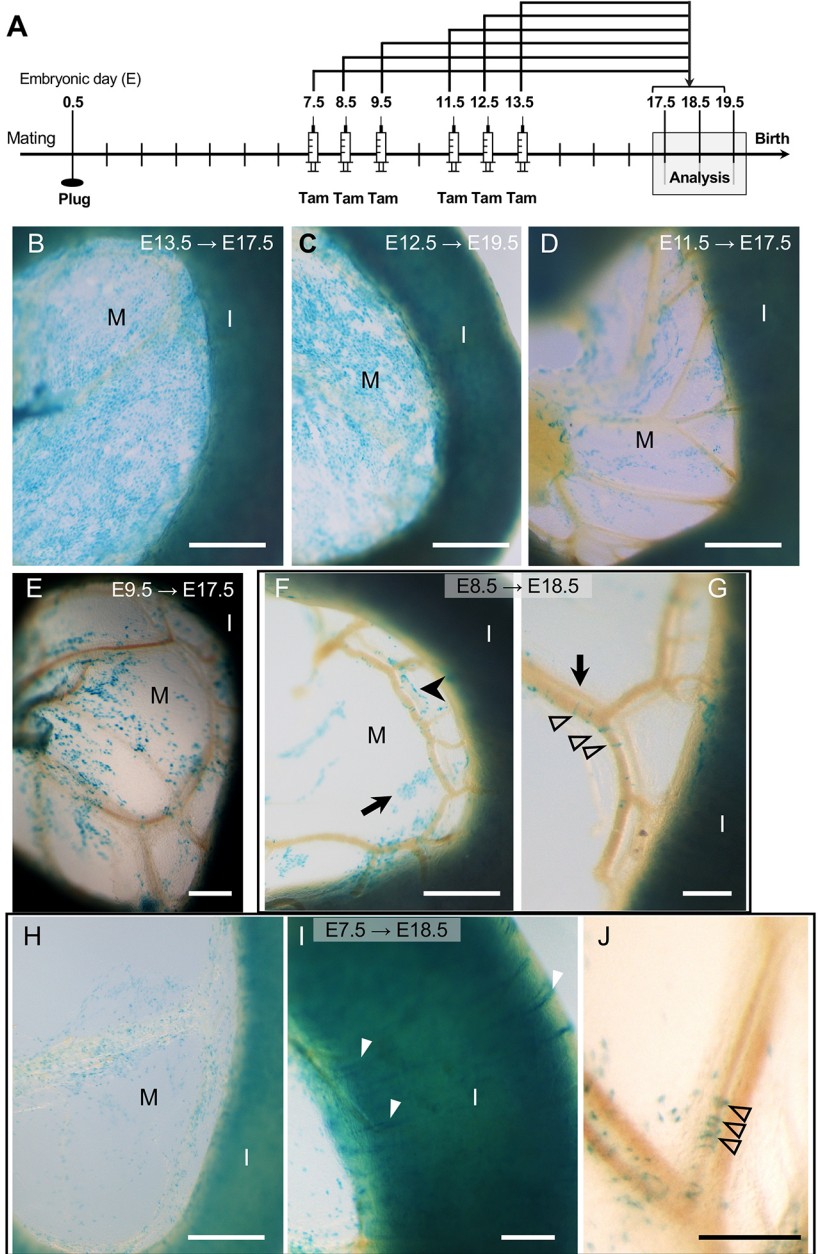

**Fig. 1. Embryonic contribution of cells of the *Wt1* lineage to visceral mesothelium or visceral and vascular smooth muscle is dependent on timing of tamoxifen dosing.** (A) Schematic of the study design: CD1 dams were time-mated with *Wt1^CreERT2/+^;Rosa26^lacZ/lacZ^* males and received tamoxifen (Tam) at the respective embryonic stages (embryonic day, E); embryos were dissected between E17.5 and E19.5, and XGal stained. (B) When Tam was given at E13.5 and embryos analysed at E17.5, the mesentery (M) and intestine (I) were well covered by XGal⁺ (*lacZ*-expressing) cells. (C,D) When Tam had been given at E12.5, the coverage of XGal-positive cells in the mesentery was less complete at E19.5. This loss of coverage of XGal-positive cells was still more pronounced E17.5 when Tam had been given at E11.5 (D). (E) Similarly, there were only patches of XGal-positive cells in the mesentery and over the intestine at E17.5 after Tam administration at E9.5. (F,G) When Tam had been given at E8.5, XGal-positive cells were observed in small patches in the mesentery (arrow, F), or in cell chains near the vasculature (arrowhead, F), or in distinct rings (unfilled arrowheads, G) surrounding the mesenteric veins (filled arrowhead, G) in the second arcade of the mesenteric vascular tree. The corresponding artery is depicted with an arrow (G). (H-J) Tam dosing at E7.5 resulted at E18.5 in XGal-positive cells scattered within the mesentery (H), in the circular musculature of the small intestine (white arrowheads, I) and surrounding a segment of mesenteric artery of the second arcade (black arrowheads, J). Scale bars: 500 µm (B-D,F,H); 200 µm (E,G,I,J).

Next, we defined a set of 15 genes each that we considered as an SMC signature (Muhl et al., 2022) and as an MC signature (Kanamori-Katayama et al., 2011) (Fig. S3G). We assessed within all *Wt1*-expressing cells the enrichment of the SMC signature genes or MC signature genes across the three developmental stages E6.5, E7.5 and E8.5, and found that at E6.5 and E8.5 the SMC gene set was expressed in a higher proportion of cells than the MC gene set, and at E8.5 the SMC gene set had a higher relative expression than the MC gene set (Fig. 3A). Correlation analysis within total *Wt1*-expressing cells of individual SMC signature or MC signature genes with *Wt1* expression across different embryonic time points revealed a positive and highly significant correlation with several genes of the respective sets (Fig. 3B,C). The correlation of *Wt1* expression with SMC genes was higher at earlier embryonic stages, but was higher with MC genes at later stages; generally, the analysis suggested a highly dynamic change in correlation across the gene sets (Fig. 3B). *Wt1*-expressing cells that significantly co-expressed

at least three genes (the threshold) of the SMC gene set were found from E6.5 in the epiblast, at E7.0 also in the primitive streak, and from E7.5 in the LPM (Fig. 3D), while *Wt1*-expressing cells that significantly co-expressed with the MC gene set above the threshold were found from E7.25 in the LPM (Fig. 3E). Next, we explored how the *Wt1*-expressing cells that expressed the SMC and/or MC gene signatures above the threshold segregated. At E7.5, *Wt1*-expressing cells separated into two distinct populations of 11 SMC signature cells and ten MC signature cells; ten of the SMC signature cells were in the LPM, while one of the MC signature cells was in the LPM (Fig. 3F,H). By contrast, at E8.5, there were 11 cells with both SMC and MC signatures above the threshold in addition to two distinct populations of 58 SMC signature cells and 33 MC signature cells (Fig. 3G). In the SMC signature group, 45 cells were localised in the LPM (78%), while 24 MC signature cells were localised in the LPM (72%) (Fig. 3I). Our analysis further revealed that *Wt1* expression levels in the SMC signature cells were generally lower

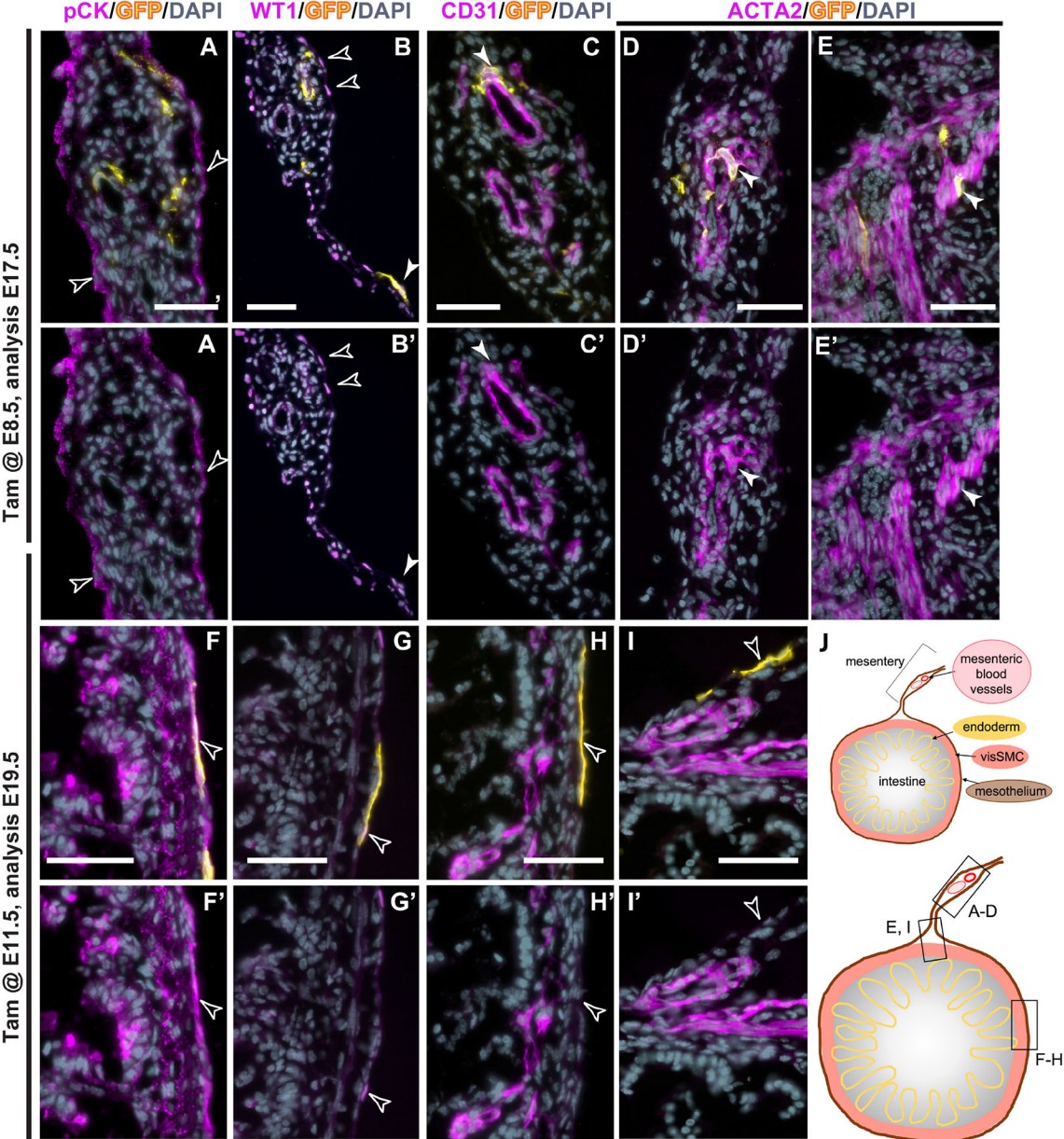

**Fig. 2. Immunofluorescence analysis of embryonic *Wt1* lineage cells in intestine and mesentery.** (A-I′) Frozen sections of intestine and mesentery of embryos after Tam dosing at E8.5 or E11.5, and dissection at E17.5 or E19.5, respectively, were analysed by immunofluorescence for co-expression of GFP with cytokeratin (A,A′,F,F′), WT1 (B,B′,G,G′), CD31 (C,C′,H,H′) and ACTA2 (D-E′,I,I′). (A-B′) After Tam administration at E8.5, co-expression of GFP with the mesothelial markers cytokeratin and WT1 was sparse (filled arrowheads), while the majority of MCs expressed no GFP (arrowheads). (C,C′) GFP-positive cells were found in close vicinity to CD31 expressed in the mesenteric vessels (arrowheads). (D-E′) At this stage, GFP was co-expressed with ACTA2 in the mesenteric vasculature (arrowheads, D,D′), and the visceral smooth muscle (arrowheads, E,E′). (F-G′) After Tam dosing at E11.5, the mesothelial markers cytokeratin and WT1 were detected in the mesothelium covering the intestine and co-expression with GFP was observed in the visceral mesothelium (arrowheads). (H-I′) Expression analysis for the vascular markers CD31 and ACTA2 showed no co-expression with GFP within the mesentery. Images are representative of six samples. (J) Schematic of the anatomical location of the different images. Scale bars: 50 µm.

than in MC signature cells, at both E7.5 and E8.5 (Fig. 3H,I, Fig. S3H,I). Out of the 11 cells with both SMC and MC gene signatures, nine were found in the LPM (Fig. S3J).

Taken together, these data suggest that a consistent population of *Wt1*-expressing cells arises from the epiblast, via the primitive streak to the emerging mesoderm and the LPM. The *Wt1*-expressing cells divided into two populations with either smooth muscle or mesothelial fate between E7.0 and E7.5, emerging predominantly in the LPM. Onset for the SMC fate in *Wt1*-expressing cells appeared to be slightly earlier and more prominent than the MC fate, with

respect to number of cells, correlation of expression and significance. Furthermore, *Wt1* expression levels were generally lower in cells with the SMC signature than in cells with the MC signature.

## Relationship between *Wt1* and *Hand1/2* expression and SMC and MC fate

The HAND1 and HAND2 transcription factors are key regulators of vascular development (Barnes et al., 2010, 2011; Vincentz et al., 2021, 2017). HAND1-expressing cells emerge from the LPM, similarly to WT1, and both HAND1 and HAND2 have been shown to

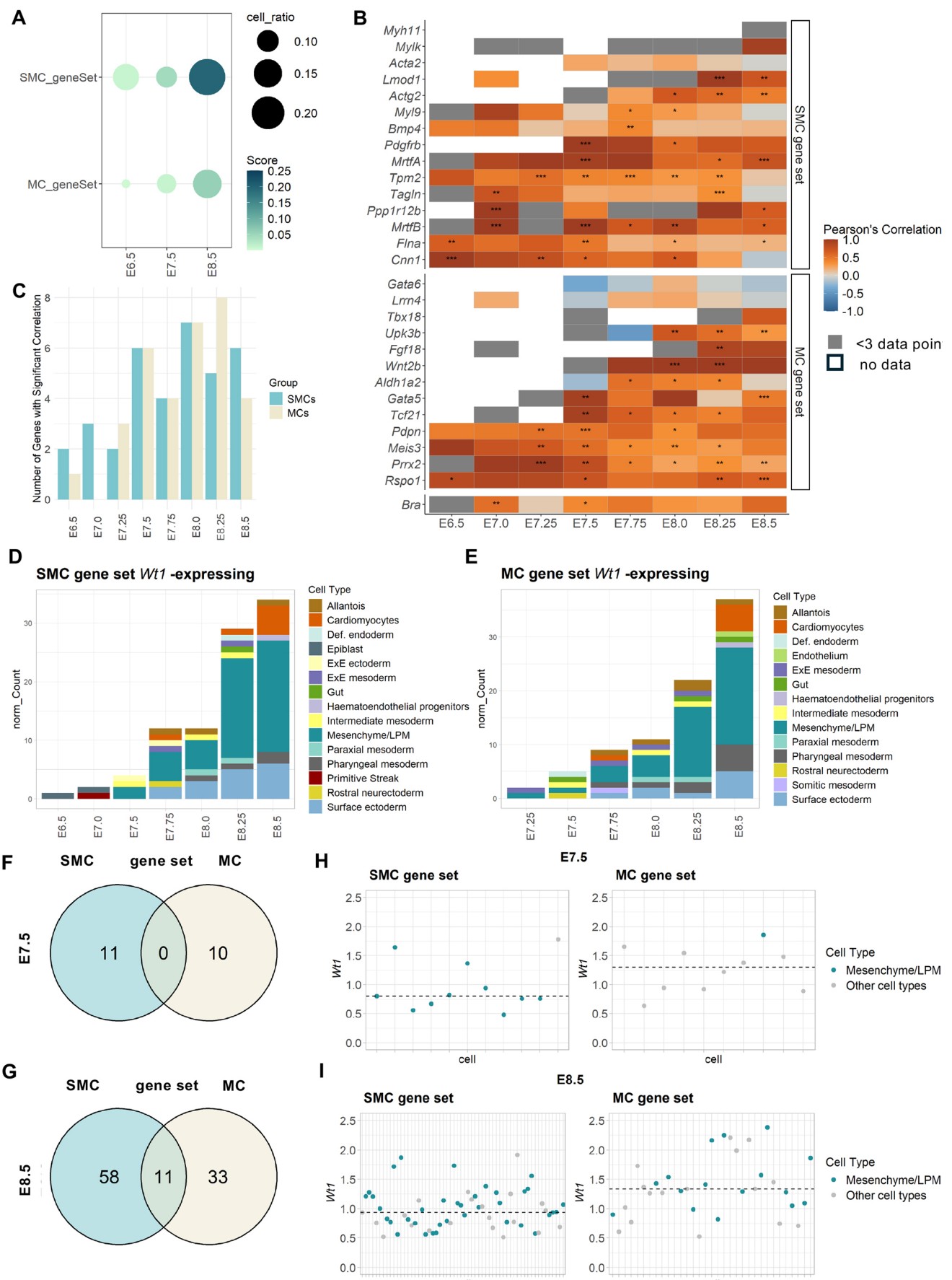

**Fig. 3.** See next page for legend.

**Fig. 3. *Wt1* expression in gastrulation-stage cells with SMC and MC signatures.** (A) Analysis of enrichment of the SMC and MC gene sets in all *Wt1*-expressing cells using AUCell. (B) Correlation plots of *Wt1* expression against SMC marker or MC signature genes in total *Wt1*-expressing cells at different embryonic time points. White boxes indicate total absence of data at the respective time point; grey boxes indicate fewer than three data points. ***$P<0.001$; **$P<0.01$; *$P<0.05$ (Pearson's correlation test). (C) Dynamic changes in the number of SMC and MC markers significantly correlated with *Wt1* expression at different embryonic stages. (D) Dynamic distribution of *Wt1*-expressing cells significantly co-expressing at least three SMC signature genes in different tissues across embryonic stages. (E) Dynamic distribution of *Wt1*-expressing cells significantly co-expressing at least three MC signature genes in different tissues across embryonic stages. Def. endoderm, definitive endoderm; EXE ectoderm, extra-embryonic ectoderm; EXE mesoderm; extra-embryonic mesoderm. (F,G) Subsets of *Wt1*-expressing cells with respective SMC or MC gene signature scoring above threshold ($n=3$) were intersected. At E7.5, *Wt1*-expressing cells separated into two distinct populations with either high SMC signature score or MC signature score. However, at E8.5, there were 11 cells with both SMC signature and MC signature score above the threshold. (H,I) *Wt1* expression levels in cells of either the SMC or MC signature groups at E7.5 and E8.5. The 11 cells expressing both SMC and MC signature genes were excluded from the E8.5 graphs. Dashed line represents median of the dataset. Other cell types in SMC gene set at E7.5 included extra-embryonic ectoderm, extra-embryonic mesoderm, pharyngeal mesoderm, rostral neuroectoderm, surface ectoderm and visceral endoderm; and in the MC gene set definitive endoderm, extra-embryonic ectoderm, mixed mesoderm, parietal endoderm, pharyngeal mesoderm, primitive streak and rostral neuroectoderm. At E8.5, other cell types in both groups included allantois, cardiomyocytes, pharyngeal mesoderm and surface ectoderm.

signify precursors of the MC and SMC lineage in mouse and zebrafish (Maska et al., 2010; Prummel et al., 2022; Reynolds et al., 2021; Yin et al., 2010). Therefore, we explored the correlation of *Hand1* or *Hand2* expression with the SMC or MC gene sets and in the context of *Wt1* expression. Total *Hand1*-expressing cells showed slightly stronger correlation and significance with the SMC gene sets from E6.5 to E7.25 than with the MC gene sets (Fig. S4A). *Hand2*-expressing cells were overall more strongly correlated with SMC and MC signature genes than were *Hand1*-expressing cells (Fig. S4B). There was significant correlation with *Wt1* expression in *Hand2*-expressing cells from E7.5 onwards, but not in *Hand1*-expressing cells (Fig. S4A,B). We further analysed the correlation of *Hand2* expression in *Wt1*-expressing cells with SMC and MC signature genes, and found limited correlation and significance with either gene set (Fig. S4C,D). The majority of *Wt1*- and *Hand2*-co-expressing cells were found in the LPM, starting from E7.0 onwards (Fig. S4E).

These results suggest that *Hand2* expression during gastrulation stages correlated with both SMC and MC fate more strongly than did *Hand1* expression. By contrast, *Hand2* expression in Wt1-expressing cells was not strongly correlated with either SMC or MC fate even though co-expressing cells were identified in the LPM.

## WT1 protein is expressed in the primitive streak and emerging mesoderm in E7.5 and E8.5 mouse embryos

Next, we determined whether WT1 protein could be detected in the tail bud of E8.5 wild-type mouse embryos using confocal IF microscopy of whole-mount stained embryos. Our analysis revealed that WT1 was expressed in emerging mesodermal cells, as well as extra-embryonic tissues, including the yolk sac and allantois (Fig. 4A-D). Co-immunostaining for BRA and confocal imaging clearly identified WT1-expressing cells in the BRA-domains of primitive streak and IM/LPM region (Fig. 4A,C,D-G; compare with Fig. S6 based on the EMAP eMouse Atlas project; Richardson et al., 2014). Co-expression analysis of individual focal planes revealed that in the most posterior region around the primitive streak WT1

expression was found in some BRA-positive cells emerging from the node (Fig. 4E), the mesenchyme including neuromesodermal progenitor cells and IM/LPM, the neural plate, primitive blood cells and allantois (Fig. 4D-G, and insets). Quantification of co-expression of WT1 and BRA (Fig. 4E,F, insets) revealed that cells with strong WT1 expression were also positive for strong BRA expression (Fig. 4H), confirming that WT1 was co-expressed in BRA-positive cells emerging from the mesoderm. Of note, our analysis of the scRNAseq dataset suggested that very few cells contained transcripts of both *Wt1* and *Bra*, whereas protein for both genes seemed to be detectable in more cells.

At E7.5, the co-expression domain between WT1 and BRA was more restricted, with WT1 mostly expressed in the epiblast, where it was co-expressed with BRA, and in the primitive streak (Fig. S7B). Further confocal image analysis of triple whole-mount IF for WT1, BRA and TAGLN in E8.5 mouse tailbuds demonstrated protein co-expression of WT1 with TAGLN in the mesoderm/paraxial mesoderm and LPM (Fig. 4I-N, Fig. S7A). These data support the observation that co-expression of WT1 and TAGLN in the emerging LPM define the smooth muscle population. Analysis of HAND2 and WT1 expression in the emerging LPM of wild-type mouse embryos at E8.5 showed that there were cells that either co-expressed these proteins, or expressed them individually in a mutually exclusive manner (Fig. S5).

Taken together, these data show that WT1 is dynamically expressed in the gastrulating embryo in cells of the primitive streak and emerging mesoderm, including the neuromesodermal progenitor cells and LPM, where it is co-expressed with HAND2 and the SMC marker TAGLN.

## Cells arising from *Wt1* lineage tracing after Tam dosing at E7.5 or E8.5 localise to three main tissues at E9.5: pro-epicardium, urogenital ridge and neural tube

To obtain a better temporal resolution of the cells that had expressed WT1 at gastrulation stages for lineage tracing of SMCs, we administered Tam at E7.5 or E8.5 and observed the range and localisation of GFP-positive cells at E9.5 (Fig. S8A).

Tam dosing at E7.5 resulted in very few GFP-positive cells, mostly found in the pro-epicardium (Fig. S8B-E). In the three litters analysed, we found that all 15 embryos had between two and 20 GFP-positive cells in the pro-epicardium, while GFP-positive cells (range 1-12 cells) in the posterior trunk region were observed only in nine out of 15 embryos (Table S2, Fig. S11). Importantly, we also detected GFP-positive cells in the neural tube of embryos after Tam at E7.5 (Fig. S8F-I) and E8.5 (Movie 1), in line with our observations in Figs 3 and 4 that WT1 was also expressed in the emerging neuroectoderm. By contrast, in embryos after Tam administration at E8.5 we observed abundant GFP-positive cells localised to the pro-epicardium at the caudal side of the forming heart (Fig. S8J-M), and to the urogenital ridge (Fig. S9B,C). We observed GFP-positive cells consistently towards the ventral midline, including the mesentery, in sections, and by using light-sheet microscopy of the trunk region of the E9.5 embryo (Figs S9B,C, S12, Movie 1, Table S3).

Together, these results demonstrate that the $Wt1^{CreERT2/+}$ lineage-tracing system detected cells that expressed WT1 at around E7.5 and E8.5 within a similar spectrum of cell progeny, but GFP-positive cell numbers were smaller when Tam was given at E7.5. This was in agreement with the scRNAseq data analysis (Fig. 3) and distribution of WT1-expressing cells by whole-mount IF (Fig. 4). Moreover, we could identify GFP-positive cells that appeared in the region of the mesentery as potential precursors for visceral and vascular smooth muscle of the developing intestinal tract.

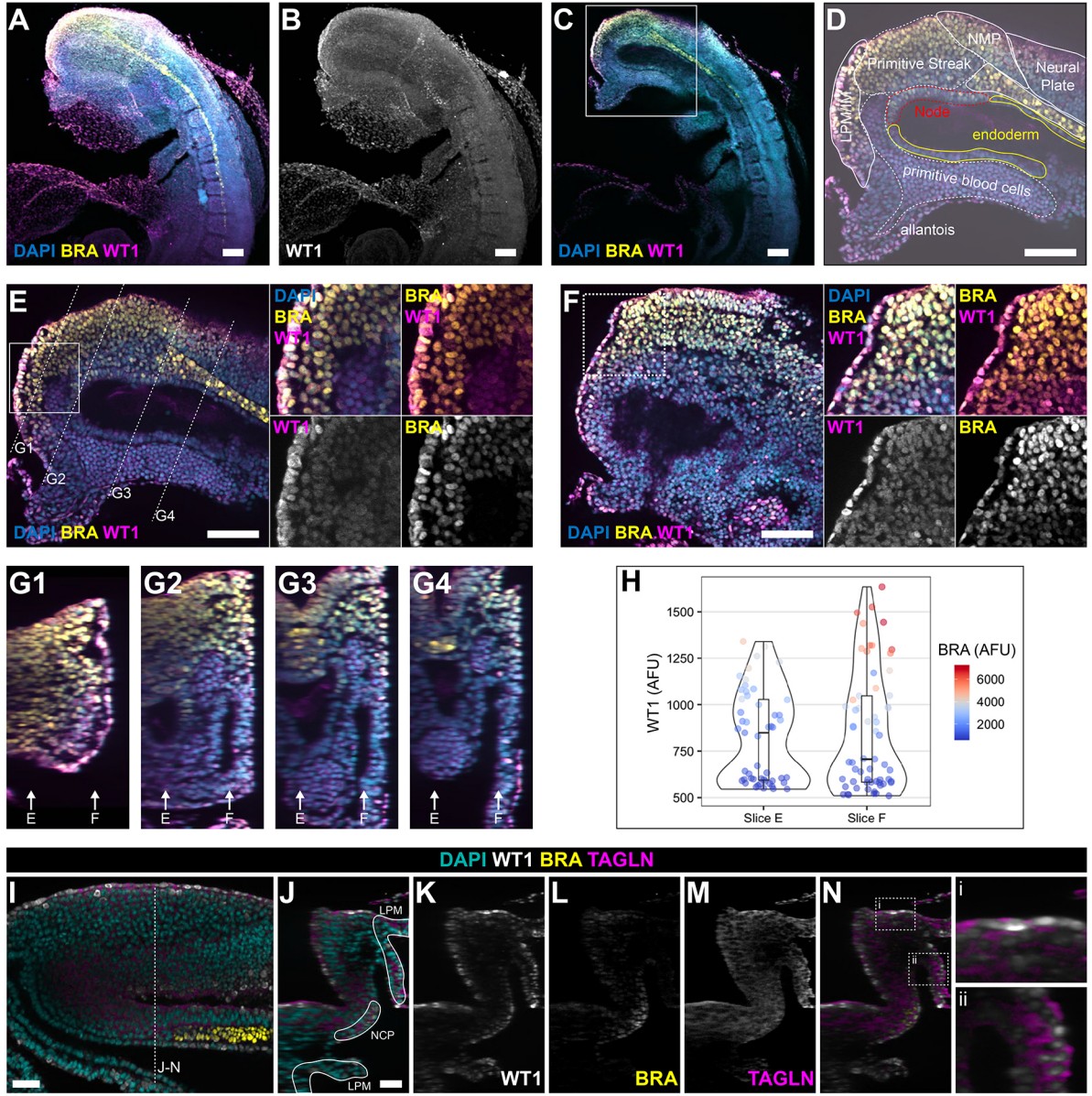

**Fig. 4. WT1 is expressed in the primitive streak, emerging mesoderm, neuromesodermal progenitor cells and LPM in E8.5 mouse embryos.**
(A-D) Whole-mount staining for WT1 and BRA in the E8.5 mouse tail bud. A and B are maximum intensity projection images generated from 150 z-sections, while C and D are focal images at sagittal level. The white box in C delineates the region shown in D,E. WT1 was expressed in the tail bud, in the allantois and yolk sac, as well as primitive streak, neuromesodermal progenitor cells and LPM/IM, where there was co-expression with BRA. In D (based on E), the different tissue domains are outlined. (E-G) Images at two focal planes through the tail bud including node, primitive streak, notochord, neuromesodermal progenitor cells, LPM/IM and neural plate, in sagittal (slice 109; E) and near-sagittal (slice 268; F) positions. Boxed regions in E and F are shown at higher magnification in insets. Lines in E indicate the position of the focal views in G1-G4. Arrows in G1-G4 indicate the position of focal planes for E and F. (H) Quantification of co-expression levels of both WT1 and BRA at the two focal slices indicated shown in E and F and indicated in G1-G4. AFU, arbitrary fluorescence units. The horizontal line is the median, the box is the interquartile range, the upper and lower whiskers correspond to the highest and lowest AFU values. n=3 embryos. (I) Whole-mount staining for WT1, BRA and TAGLN in the E8.5 mouse tail bud. Oblique focal plane is shown; dotted line indicates the focal plane for the images in J-N. (J) LPM and notochordal plate (NCP) are outlined in a transverse focal plane image. (K-M) Expression of WT1 (K), BRA (L) and TAGLN (M). WT1 and TAGLN were co-expressed in the surface cells covering the paraxial mesoderm and in the LPM. (N) Merged image of the staining shown in K and M, with WT1 expression in white and TAGLN expression in purple to highlight regions of co-expression. Scale bars: 100 μm (A-F); 50 μm (I-N).

## Cells of the *Wt1* lineage contribute to the development of the vascular and visceral smooth muscle between E10.5 and E15.5

To elucidate the developmental path and fate of the vSMCs and visSMCs using the emergence of the GFP-positive cells, we analysed embryos at E10.5 after Tam dosing at E8.5. At this stage, GFP-positive cells were predominantly seen in the region of the urogenital ridge and the heart (Fig. S9A,D-F′). In tissue sections, GFP-positive

cells localised in the region between the dorsal aorta and the ventral mesentery, and within the urogenital ridge (Fig. S9E,E′). Some of the cells within the region between dorsal aorta and ventral mesentery appeared to be situated close to endothelial cells (Fig. S9F,F′).

In embryos dosed with Tam at E8.5 and analysed at E12.5 (Fig. 5A), we found GFP-positive cells in a spotty appearance within the mesentery of the developing small intestinal segment of the intestinal tract (Fig. 5B,B′). IF analysis on sections through this

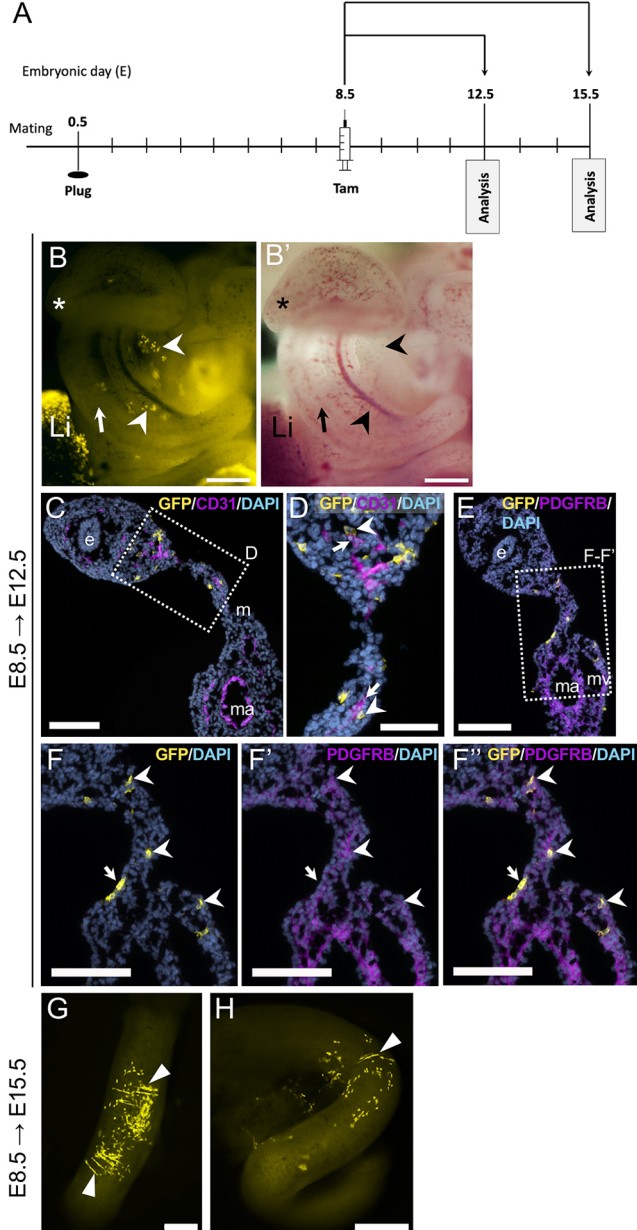

**Fig. 5. Contribution of *Wt1* lineage cells to the mesentery near developing vasculature and to the intestinal wall.** (A) Schematic of the study design: embryos were analysed at E12.5 or E15.5 after timed mating of *Wt1^CreERT2/+^*;*Rosa26^mTmG/mTmG^* males with CD1 females and Tam dosing at E8.5. (B,B′) View of the intestinal loop and mesentery. GFP-positive cells (arrowheads) were found within the mesentery of the developing intestinal tract (arrows; asterisks mark the caecum). Note the abundant GFP fluorescence in the liver (Li). (C,D) Immunofluorescence of CD31 and GFP on sections through this region showed GFP-positive cells within the mesentery and in the intestinal wall, near the developing microvascular plexus (C). Higher magnification (D) reveals that GFP-positive cells (arrowheads) are often arranged close to neighbouring endothelial cells (arrows) of the vascular plexus in the intestinal wall and mesentery. (E-F″) Immunofluorescence for PDGFRB and GFP on neighbouring sections showed that GFP-positive cells within the intestinal wall mesenchyme and mesentery co-expressed PDGFRB (arrowheads), while a few mesothelial GFP-positive cells were PDGFRB negative (F-F″, arrows). (G,H) Representative images from two different embryos at E15.5 after Tam dosing at E8.5. GFP-positive cells (arrowheads) were found within a segment of the small intestine in a circular arrangement. e, endoderm; m, mesentery; ma, mesenteric artery; mv, mesenteric vein. Scale bars: 500 µm (G,H); 300 µm (B,B′); 100 µm (C,E,F-F″); 50 µm (D).

region revealed that GFP-positive cells were localised within the mesentery and in the developing intestinal wall near the forming vascular plexus (Fig. 5C,D). GFP-expressing cells were distinct from endothelial cells but closely associated (Fig. 5D). Furthermore, our analysis showed that some of the GFP-labelled cells within the mesentery and the intestinal wall co-expressed PDGFRB, suggesting that they had started the smooth muscle differentiation programme (Fig. 5E-F″). Occasionally, we found a few GFP-positive cells in the mesothelial covering of the mesentery (Fig. 5E,F). Together, these results indicate that *Wt1* lineage-traced cells, labelled at around E8.5, appeared in the vicinity of the developing vasculature and co-expressed PDGFRB, a key marker of vascular smooth muscle progenitor cells during stages of intestinal patterning and angiogenesis. When endpoint analysis was performed at E15.5, we detected GFP-positive cells within the visceral smooth muscle and surrounding mesenteric vessels (Fig. 5G,H), similar to findings from the endpoint analysis at around E18.5 (Fig. 1I). We also explored the *Wt1* lineage in embryos after Tam administration at E6.5 and analysis at E14.5, but observed only very few GFP-labelled cells near the intestine and gonad in one out of five analysed embryos (from three litters; Fig. S10).

## Tam administration to ablate *Wt1* at E7.5 leads to disorganised developing vasculature in the intestine

We next examined how deletion of *Wt1* during gastrulation stages would affect vascularisation of the developing intestine and mesentery by administering Tam at E7.5 to three litters of *Wt1^CreERT2/fl^*; *Rosa26^mTmG/mTmG^* (KO@E7.5) and *Wt1^fl/+^*;*Rosa26^mTmG/mTmG^* (control) embryos (Fig. 6A). Cre activity could be demonstrated in the KO@E7.5 embryos by virtue of GFP expression throughout the embryos, most prominently in the heart (Fig. 6B), suggesting that WT1 had been inactivated in at least some of the cells expressing *Wt1* at E7.5. In the intestines of KO@E7.5 embryos, we found higher variability in the density of endothelial CD31 expression, and in the vascular network formation, when compared to control embryos (Fig. 6C-J). Images captured by 3D confocal microscopy were tested for batch effects, which could be neglected (Fig. 6F). Our analysis revealed that the variability in the density of CD31 expression was higher in the KO@E7.5 group than in the controls (Fig. 6E,G), and the coefficient of variation for most parameters of vascular network formation was much higher in the KO@E7.5 embryos than in controls (Fig. 6J), again revealing higher variability. Furthermore, we could detect ACTA2-expressing cells around CD31-expressing vessels within the developing intestine, and observed that in the KO@E7.5 embryos the ACTA2-expressing cells appeared less aligned and attached to the endothelial cells than in the controls (Fig. 6K).

Collectively, these data indicate that Tam administration led to disruption of WT1 function in cells that expressed *Wt1* at around E7.5, affecting vessel formation in the developing intestine and mesentery at later stages, in both endothelial and mural cells.

## DISCUSSION

In this study, we have for the first time determined the temporal and spatial relationship between the emerging intestinal vascular and visceral smooth muscle cells and the expression of the transcriptional regulator WT1 during gastrulation and post-gastrulation stages of mouse embryonic development. Our analysis revealed that WT1 is expressed in a progenitor cell pool in the emerging mesoderm of the gastrulating embryo. Some of these cells differentiate into smooth muscle cells in the intestine and mesentery. Our data indicate that this process is temporally independent of the formation of the mesothelium covering these tissues, which also expresses WT1.

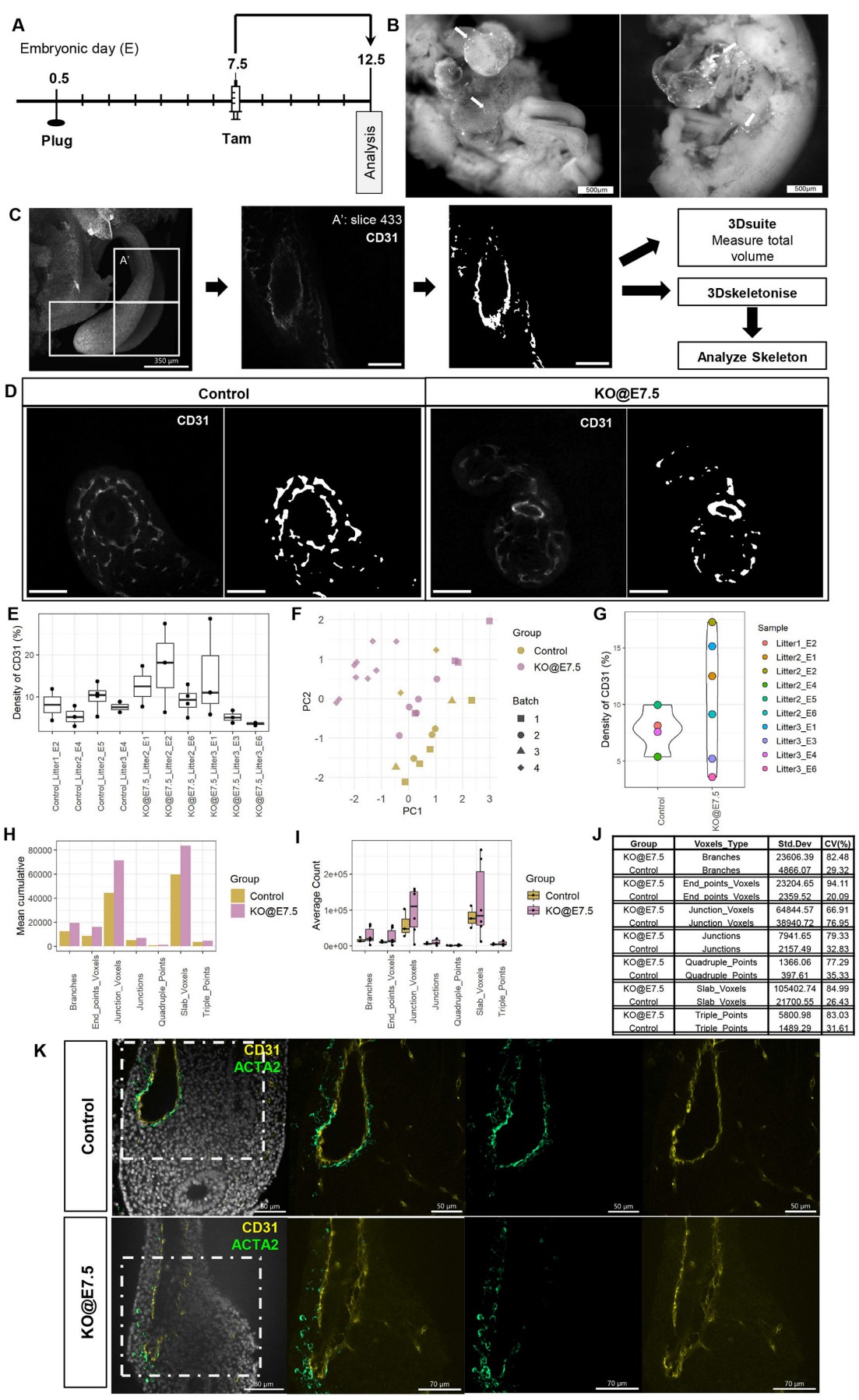

**Fig. 6.** See next page for legend.

**Fig. 6. Effects on intestinal vasculature development in *Wt1^CreERT2/fl^*; *Rosa26^mTmG/mTmG^* embryos after Tam dosing.** (A) Schematic of the study design: after dosing with Tam at E7.5, *Wt1^CreERT2/fl^;Rosa26^mTmG/mTmG^* (KO@E7.5) and *Wt1^+/fl^;Rosa26^mTmG/mTmG^* (control) embryos were analysed at E12.5. (B) Ventral/lateral views of GFP fluorescence in two KO@E7.5 E12.5 embryos after removal of the ventral body wall and lateral tissues. A number of GFP-positive cells are seen over the heart, lungs and liver (arrows), indicating Cre recombination activity. (C) Workflow of whole-mount CD31 staining analysis. Regions of interest (ROIs; white boxes) were processed to measure the total volume (3Dsuite) and the branching characteristics (AnalyzeSkeleton) of the vasculature of the respective regions. (D) Selected focal planes showing CD31 signals of a KO@E7.5 and a control embryo. (E) Overall distribution of percentage CD31 density in different ROIs of each embryo. In the graph, the horizontal line is the median, the box is the interquartile range, the upper and lower whiskers correspond to the highest and lowest percentage values. (F) Principal component analysis plot of percentage CD31 density of the ROIs in each embryo processed in different batches. (G) Distribution of mean percentage CD31 density in KO@E7.5 and littermate controls. Distribution in the control group was consistently lower (s.d.=1.89; coefficient of variation=24.36%), while the distribution in KO@E7.5 embryos was more widely spread throughout (s.d.=5.46; coefficient of variation=50.65%). (H) Mean cumulative count of different voxel types. (I,J) Distribution of samples in each group at different voxel types. Branching characteristics revealed a higher variability and coefficient of variation in KO@E7.5 compared to control embryos. In the graph, the horizontal line is the median, the box is the interquartile range, the upper and lower whiskers correspond to the highest and lowest average counts. (K) Analysis of focal planes of *z*-stacks after whole-mount IF for CD31 and ACTA2 revealed that SMC interactions with endothelial cells were affected in KO@E7.5 compared to control embryos. Boxed areas are shown at higher magnification on the right. Scale bars: 500 µm (B); 350 µm (C, left); 100 µm (C, right; D); 80 µm (K, left lower image); 70 µm (K, right lower images); 60 µm (K, left upper image); 50 µm (K, right upper images).

Thus, our data shed new light onto smooth muscle development as a process independent of mesothelium development, and the temporospatial involvement of WT1 in this process (Fig. 7).

Using Tam-dependent, inducible, genetic lineage tracing, we analysed the embryonic contribution of *Wt1*-expressing cells

to visSMCs and vSMCs of the intestine. In previous studies, we had shown that *Wt1*-expessing cells give rise to vSMCs in the mesentery and intestine using a non-inducible *Wt1*-based lineage-tracing system (Wilm et al., 2005). Similarly, by using the same or an alternative *Wt1*-driven genetic reporter system, it had been demonstrated that *Wt1*-expressing progenitor cells give rise to visSMCs and vSMCs not only in the intestine but also in the lungs (Cano et al., 2013; Carmona et al., 2013; Que et al., 2008); however, the temporal involvement and spatial significance of *Wt1* expression during smooth muscle specification and differentiation had not been determined. In a follow-on study, we determined that *Wt1*-expressing cells in postnatal stages fail to give rise to intestinal visSMCs and vSMCs, by making use of the same Tam-dependent temporal activation process for lineage analysis as utilised here (Wilm et al., 2021). We concluded that the source of intestinal and mesenteric SMCs had to be cells expressing *Wt1* during embryonic development; however, it was not clear whether these cells arise from the overlying mesothelium, which also expresses WT1 during embryonic development, as had been our previous hypothesis (Wilm et al., 2005).

To address this question, we administered Tam between E7.5 and E13.5 using the *Wt1^CreERT2/+^* driver in combination with either the *Rosa26^lacZ^* or *Rosa26^mTmG^* reporter systems, and analysed the presence of *lacZ*-expressing and GFP-labelled cells in foetuses just before birth (E17.5, E18.5 or E19.5). Our analysis showed that Tam dosing at E7.5 or E8.5 resulted in labelling of vSMC and visSMCs, which had originated from *Wt1*-expressing cells. When Tam was given at E9.5 or later stages, the labelled cells were confined to the mesothelial covering of the mesentery and intestine. This finding demonstrates a clearly defined mechanism for the emergence of the mesothelial lineage that is separate from that of the vascular/visceral smooth muscle lineage, indicating a role for WT1 in distinct mechanisms.

Other studies have previously used the Tam-inducible *Wt1^CreERT2/+^* driver in combination with reporter systems to elucidate the lineage

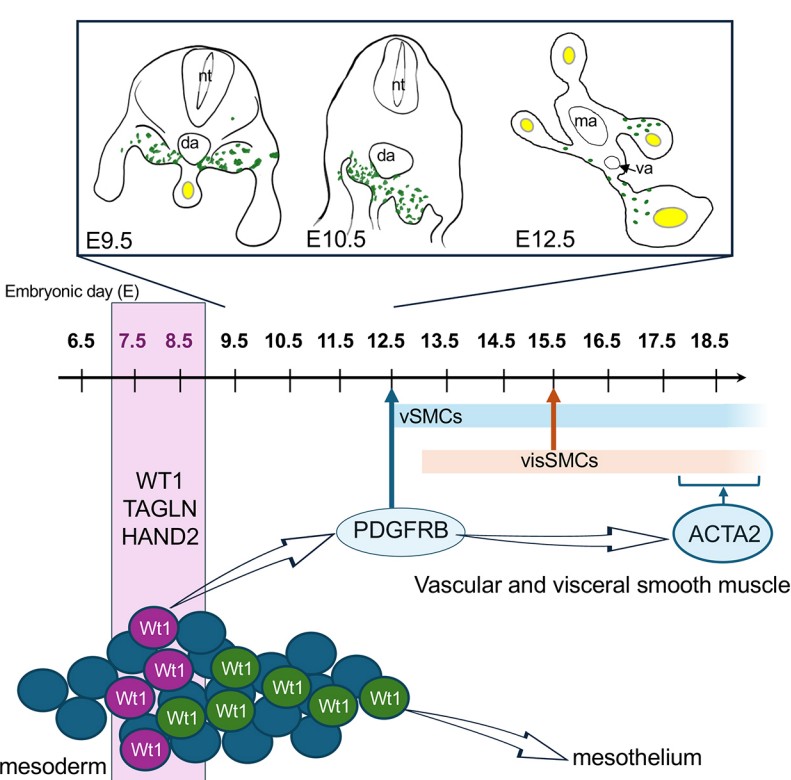

**Fig. 7. The embryonic lineage contribution of WT1-expressing cells to visceral and vascular smooth muscle cells is uncoupled from the mesothelial lineage.** Lineage tracing of GFP-labelled cells via the *Wt1^CreERT2/+^*; *Rosa26^mTmG/mTmG^* system after Tam at E7.5 or E8.5 demonstrated that cells appeared at the dorsal base of the mesentery at E10.5. At E12.5 GFP-positive cells were closely aligned to the forming endothelial plexus and co-expressed PDGFRB in the mesentery of the midgut. At E15.5, GFP-positive visceral smooth muscle cells were detectable in the foetal intestine. At around E18.5, GFP-positive vSMCs and visSMCs co-expressed ACTA2. By contrast, GFP-labelled mesothelial cells emerged in lineage-traced embryos predominantly when Tam had been given from E9.5 onwards. da, dorsal aorta; ma, mesenteric artery; va, vitelline artery.

and origin of smooth muscle cells in the lungs. Contribution to airway smooth muscle, vascular smooth muscle and parenchyma was observed in late-stage mouse foetuses when Tam had been given between E9.5 and E11.5 (Dixit et al., 2013; Moiseenko et al., 2017). Time-lapse experiments of *ex vivo* cultured lungs from lineage-traced E12.5 embryos after Tam dosing at E10.5 revealed a direct contribution of mesothelial cells towards the deeper lung parenchyma (Dixit et al., 2013). These results suggest that the smooth muscle of the airways and lung vasculature may have a contribution via the mesothelium at later developmental stages, in a mechanism different to that described here for the intestinal tract.

In the heart, WT1-based lineage tracing has demonstrated a contribution of epicardial cells to coronary vessel formation (Quijada et al., 2020; Rudat and Kispert, 2012; Wilm et al., 2005; Zhou et al., 2008). Based on *in situ* hybridisation and IF on sections of E8.5 embryos, WT1 is not expressed in the developing heart at this stage. By contrast, non-inducible $Wt1^{CreEGFP}$;$Rosa26^{mTmG}$ embryos at E8.5 have a number of GFP-expressing cells in the forming heart, indicating an early contribution of Wt1-expressing cells (Rudat and Kispert, 2012). However, WT1 is also expressed in non-epicardial cells in the heart, including myocardium, which makes interpretation of lineage experiments more challenging (Rudat and Kispert, 2012). In addition, our own postnatal lineage-tracing experiments using the Tam-inducible $Wt1^{CreERT2/+}$ driver had shown that after birth Wt1-expressing cells contribute to coronary SMCs (Wilm et al., 2021), suggesting a different mechanism of coronary angiogenesis and maintenance of coronary vasculature than in the intestinal tract and mesentery.

The Tam-inducible lineage-tracing system is dependent on Tam-driven recombination activity, which then irreversibly switches on the reporter gene expression. Several studies have analysed the temporal delay between Tam administration and reporter expression, indicating that, depending on route of administration (oral gavage or intraperitoneal injection), a temporal delay of 12-18 h could be observed in other genetic systems (Nguyen et al., 2009; Zhu et al., 2008). This temporal delay needs to be factored in when interpreting the results of the current study, suggesting that about 0.5 embryonic days may have to be added to the stages indicated for the recombination initiation.

The Tam-inducible lineage-tracing system is based on the $Wt1^{CreERT2/+}$ driver. We and others have previously shown that the $Wt1^{CreERT2/+}$ driver fails to provide 100% recombination efficiency, leading to inefficient cell labelling of between 14.5% and 80% (Chen et al., 2014; Li et al., 2013; Wilm et al., 2021). A range of reasons could contribute to this observation, including inefficiency of the recombination event and/or insufficiency of Tam delivery and physiological distribution, as discussed previously (Wilm et al., 2021). These deficiencies inherent to the Tam- and Cre-driven lineage-tracing/gene-ablation system need to be taken into account when interpreting the distribution and coverage of labelled cells and functional disruption of genes within tissues.

One of the surprising findings of this study emerged from our analysis of scRNAseq data (Pijuan-Sala et al., 2019) showing that, from E6.5 onwards, *Wt1* is expressed in the epiblast, primitive streak and emerging mesoderm, and in the LPM where it is co-expressed with key markers of SMC development. Analysis of WT1 expression in early-stage mouse embryos had been previously performed by *in situ* hybridisation showing *Wt1* transcripts in the intermediate mesoderm/LPM in E9.0 mouse embryos (Armstrong et al., 1993). Because further insights into *Wt1* expression or function at gastrulation stages have not been published yet, our results are the first to shed light onto WT1 gastrulation-stage expression or engagement.

We have identified *Wt1* expression in epiblast and primitive streak cells in embryos up to about E7.25, and subsequently a shift to specifically identified mesodermal cells in embryos from E7.5 to E8.5 where they are predominantly present in the LPM. *Wt1*-expressing cells with an SMC signature emerged slightly earlier and more robustly from E6.5 onwards than *Wt1*-expressing cells with a MC signature; however, interestingly, there were a few *Wt1*-expressing cells that have both SMC and MC signatures. We confirmed co-expression of WT1 protein with BRA in the primitive streak/emerging mesoderm, and with the key SM marker TAGLN in the LPM.

These results demonstrate that WT1-expressing cells unexpectedly emerge already in gastrulation-stage embryos. Our findings indicate that the SMCs labelled by *Wt1* lineage tracing are not derived from *Wt1*-expressing MCs arising first and then differentiating into visSMCs and vSMCs. Instead, the slightly earlier onset and prevalence of the *Wt1*-expressing SMC signature cells during gastrulation appears to provide a direct link to the labelled SMCs many days later in the embryo.

The peritoneal mesothelium emerges due to separation of the LPM into somatic and splanchnic mesoderm, which surround the peritoneal cavity; the splanchnic mesoderm forms the mesodermal component of the intestinal tract (Funayama et al., 1999). Detailed analysis of the emergence of the mesothelium has been provided in chick and quail embryo models (Thomason et al., 2012; Winters et al., 2012), demonstrating that the bilateral splanchnopleure, consisting of splanchnic mesoderm and endoderm, turns into a trilaminar tissue with a mesenchymal layer emerging between an outer mesodermal epithelium and the endoderm. In the chick embryo, the outer epithelium is situated on a basal lamina already at Hamburger–Hamilton stage (HH) 13 (~19 somites, 48 h), whereas cytokeratin expression, as indicator for epithelium, starts only at HH19 (~40 somites, 71 h; roughly corresponding to mouse embryos at Theiler stage 17 or E10.5), and WT1 is not expressed at this stage yet (Winters et al., 2012). The outer epithelium is defined as mesothelium only from HH29 (6 days) onwards (chick and quail), when it has organised into a simple squamous epithelium (Thomason et al., 2012; Winters et al., 2012).

In zebrafish, *hand2* expression has been presented as a marker for mesothelial progenitor cells emerging from the lateral most region of the LPM at tailbud stage (i.e. at late gastrulation stages) (Prummel et al., 2022). Co-expression with Tagln in the LPM-emerging cells, and a more limited expression of *wt1a/b*, restricted to the cells of the anterior visceral peritoneum, and absent posteriorly and in the parietal mesothelium, suggested that *hand2* was a more encompassing marker for arising mesothelium. Furthermore, *hand2* mutant zebrafish had disorganised LPM-derived cells, in particular a lack of parietal mesothelium, indicating an upstream regulatory role of *hand2* in mesothelium formation (Prummel et al., 2022).

In the mouse embryo, the peritoneal mesothelium has been shown to arise between E10.5 and E11.5 since at E11.5, a continuous layer of cells expressing cytokeratin and WT1 is present (Wilm et al., 2005). Further information on mesothelial precursor populations and lineage in mouse embryonic development is limited: non-inducible *Hand2*- or *Hand1*-based lineage tracing showed labelling of mesothelium over the liver in E14.5 or E15.5 mouse embryos but no temporal analysis (Prummel et al., 2022). A non-inducible *Hand1* lineage has also been shown to label the visceral smooth muscle in mouse embryos at E14.5 (Barnes et al., 2010), and *Hand1* deletion in the LPM affected the formation of the visceral smooth muscle (Maska et al., 2010).

Our analysis of the correlation of SMC or MC signature gene expression in *Hand1*- or *Hand2*-expressing cells during gastrulation, in

comparison to that in *Wt1*-expressing cells, revealed that *Hand2* expression is more highly correlated with both SMC and MC signature gene expression than is *Hand1*, and overlaps with *Wt1*-expressing cells. In addition, both HAND2 and WT1 are present in an overlapping domain in the posterior LPM in the E8.5 embryo. These data indicate that during gastrulation and in the emerging mesoderm *Wt1*- and *Hand2*-expressing cells have similar expression profiles and WT1 and HAND2 similar expression domains.

Visceral smooth muscle arises from around E13 in the mouse embryo based on ACTA2 IF (Walton et al., 2016; Wilm et al., 2005), while the vascular smooth muscle in the intestine and mesentery can be detected from E12.5 during angiogenic remodelling, with mature vessels covered by mural cells, detectable from E15.5 onwards (Hatch and Mukouyama, 2015; Wilm et al., 2005). A complex signalling network including the myocardin/serum response factor (SRF) interaction controls SMC differentiation by regulating expression of cytoskeletal smooth muscle proteins, including ACTA2, TAGLN and smooth muscle myosin heavy chain. Furthermore, PDGFRB signalling regulates vSMC proliferation, migration and differentiation (Donadon and Santoro, 2021). In this study, we have used a signature of the key regulators and markers of SMC fate and differentiation to determine their co-expression with *Wt1* already from E7.5 onwards in the LPM. In E12.5 embryos, *Wt1* lineage-traced cells in the developing intestinal/mesenteric vascular network co-expressed PDGFRB after activation of the reporter at around E8.5. Importantly, Tam-induced knock out of *Wt1* at E7.5 affected vascularisation of the intestine and mesentery in E12.5 mutant embryos compared to controls: the endothelial network was more irregular based on the higher variability in cell density and branching, and the forming mural/vSMCs were less aligned with the vascular networks. These results strongly indicate a role for WT1 in smooth muscle development, originating in cells of the emerging mesoderm. Furthermore, our analysis of *Wt1* lineage tracing at earlier stages (E9.5, E10.5) of development suggest that the *Wt1* lineage-traced cells originating at E8.5 may migrate into the mesentery and towards the developing vascular network.

Taken together, our data clearly demonstrate that the visSMCs and vSMCs that arise from WT1-expressing cells at around E7.5 or E8.5 are independent in their specification and differentiation from the mesothelial cells of the peritoneum. These can only be tagged after initiation of WT1-controlled recombination from around E9.5 onwards. Our results therefore indicate an early emergence of the visSMC and vSMC lineage linked to WT1 expression in the gastrulating embryo, and an uncoupling from the mesothelial lineage. A detailed mechanistic role of WT1 in both processes remains to be elucidated.

## MATERIALS AND METHODS
### Mice
The following compound mutants were used in this study and bred in-house: $Wt1^{CreERT2/+};Rosa26^{lacZ/lacZ}$ [$Wt1^{tm2(cre/ERT2)Wtp}$;$Gt(ROSA)26Sor/J$] (Soriano, 1999; Zhou et al., 2008), $Wt1^{CreERT2/+}$;$Rosa26^{mTmG/+}$ [$Wt1^{tm2(cre/ERT2)Wtp}$;$Gt(ROSA)26Sor^{tm4(ACTB-tdTomato,-EGFP)Luo}/J$] (Muzumdar et al., 2007) and $Wt1^{CreERT2/fl}$;$Rosa26^{mTmG/+}$ [$Wt1^{tm2(cre/ERT2)Wtp}$;$Wt1^{fl}$;$Gt(ROSA)26Sor^{tm4(ACTB-tdTomato,-EGFP)Luo}/J$] (Martínez-Estrada et al., 2010). CD1 female mice (at least 10 weeks old, Charles River, Harlow, UK) were time-mated with $Wt1^{CreERT2/+}$;$Rosa26^{lacZ/lacZ}$ or $Wt1^{CreERT2/+}$;$Rosa26^{mTmG/+}$ males, and noon of the day on which a vaginal plug was detected was considered as E0.5. CD1 female mice time-mated by Charles River were ordered in. Mice were housed in individually ventilated cages under a 12-h light/dark cycle, with *ad libitum* access to standard food and water. All animal experiments were performed under a Home Office licence granted under the UK Animals (Scientific Procedures) Act 1986 and were approved by the

University of Liverpool AWERB committee. Experiments reported are in line with the ARRIVE guidelines.

### Tam dosing
For *Wt1* lineage tracing in embryos, pregnant CD1 mice after mating with $Wt1^{CreERT2/+};Rosa26^{lacZ/lacZ}$ or $Wt1^{CreERT2/+};Rosa26^{mTmG/+}$ male mice were dosed with 100 µg/g body weight of Tam [T5648, Sigma-Aldrich; 40 mg/ml, in corn oil (C8267, Sigma-Aldrich)] via oral gavage once on various days and analysed between E9.5 and E19.5. Pregnant dams were monitored for their wellbeing including daily weight checks. For the *Wt1* knockout, pregnant $Wt1^{fl/fl};Rosa26^{mTmG/mTmG}$ female mice mated with $Wt1^{CreERT2/+};Rosa26^{mTmG/mTmG}$ male mice were given 100 µg/g body weight of Tam via oral gavage at E7.5.

### XGal staining
Dissected embryos or tissues were fixed in 2% paraformaldehyde (PFA)/0.25% glutaraldehyde in PBS for 1-1.5 h. Whole-mount XGal staining was performed overnight by incubating the embryos/tissues in XGal solution [5 mM potassium hexacyanoferrate(II), 5 mM potassium hexacyanoferrate(III), 1× PBS, 2 mM $MgCl_2$, 0.02% NP-40, 0.01% sodium deoxycholate, 1 mg/ml XGal (5-bromo-4-chloro-3-indolyl-beta-D-galactopyranoside dissolved in N,N dimethyl formamide) in the dark, followed by overnight post-fixation in 4% PFA (in PBS) at 4°C (Wilm et al., 2005; 2021). All reagents were from Sigma-Aldrich.

### Immunofluorescence
The following primary antibodies were used: anti-WT1 rabbit polyclonal (1:200 to 1:500; clone C-19, sc-192, Santa Cruz Biotechnology), anti-WT1 mouse monoclonal (1:50; clone 6F-H2, M3561, Dako), anti-ACTA2 mouse monoclonal (1:100 to 1:200; clone 1A4, A2547, Sigma-Aldrich), anti-ACTA2 rabbit polyclonal (1:200; ab5694, Abcam), anti-Pecam/CD31 rat monoclonal (1:50 to 1:200; 550274, BD Pharmingen), anti-GFP rabbit or goat polyclonal (1:5000; ab6556 or ab6673, Abcam) (Wilm et al., 2021), anti-brachyury goat polyclonal (1:200; PA5-46984, Invitrogen/Thermo Fisher Scientific), anti-PDGFRB rat monoclonal (1:100; clone APB5, eBioscience), anti-TAGLN/SM22 mouse monoclonal (1:100; 60213-1-Ig, Proteintech). The anti-ACTA2 and anti-TAGLN/SM22 antibodies were directly labelled using the Zenon direct labelling kit (Zenon™ Alexa Fluor® 647 Conjugation Kit - Lightning-Link®, Z24107, Invitrogen/Thermo Fisher Scientific) according to the manufacturer's instructions. Secondary antibodies were Alexa fluorophore-coupled (donkey anti-rabbit 488, A-21206; donkey anti-goat 594, A-11058; donkey anti-rat 594, A-21209; Invitrogen/Thermo Fisher Scientific) and were used at a dilution of 1:1000. DAPI (D9542, Sigma-Aldrich) was used at 1:250-1:1000 for nuclear counterstaining.

### Frozen sections
Tissues were fixed in 4% PFA for 30-90 min, protected in 30% sucrose overnight, placed in Cryomatrix (Thermo Fisher Scientific) and snap-frozen. Frozen sections were generated at 8-10 µm on a Thermo Scientific HM525 NX cryostat. IF staining was performed following standard protocols (Wilm et al., 2005). Two bleaching steps of 20 min each in 3% $H_2O_2$ in methanol were included for embryos or their tissues carrying the $Rosa26^{mTmG/+}$ mutation in order to remove the tdTomato fluorescence (Lua et al., 2015). Subsequently to completed IF, sections were coverslipped with Fluoro-Gel (with Tris buffer; Electron Microscopy Sciences), sealed and stored in the dark at 4°C.

### Whole tissues and embryos
Whole tissues/embryos were fixed in 4% PFA for 60 min on ice and stored in 0.0025% Triton X-100 (Sigma-Aldrich) in PBS at 4°C until used. Tissues/embryos were permeabilised using 0.25% Triton X-100 (PBS) for 60 min on ice. IF staining was performed using previously published protocols (Baillie-Johnson et al., 2015; Beccari et al., 2018; Turner et al., 2017). E7.5 or E8.5 IF-stained embryos were mounted on coverslips within SecureSeal imaging spacers (Grace Bio-Labs) and embedded in RapiClear 1.52 solution (SunJin Lab) overnight. E12.5 IF-stained embryos were mounted on coverslips within 1.0 mm iSpacer (SunJin Lab) in RapiClear 1.52 solution for at least two nights. Samples were stored in the dark at 4°C before imaging.

## Microscopy

### Epifluorescence microscopy

IF sections were imaged on a Leica DM 2500 upright microscope with a Leica DFC350 FX digital camera and a Leica EL6000 fluorescence light source supported by the Leica Application Suite software package (LAS, version 3 or 4; Leica Microsystems).

### Whole-mount microscopy

XGal-stained and GFP-positive embryos and tissues were imaged using a Leica MZ 16F dissecting microscope equipped with a Leica DFC420 C digital camera and a Leica EL6000 fluorescence light source supported by LAS software.

### Confocal microscopy

Imaging of whole-mount embryos was performed using an Andor Dragonfly spinning disc confocal microscope (Andor, Oxford Instruments) mounted on an inverted Leica DMi8 base using either a 10× (0.45NA) or a 25× water-dipping objective (NA0.95). The z-steps for the 10× and 25× objectives were 1.32 µm and 0.35 µm, respectively. DAPI, Alexa 488, 568 and 647 were sequentially excited using 405 nm, 488 nm, 561 nm and 637 nm laser diodes, respectively, and emitted light reflected through 450/50 nm, 525/50 nm, 600/50 nm and 700/25 nm bandpass filters, respectively. An iXon 888 EM-CCD camera was used to collect emitted light, and data were captured using Fusion version 5.5 (Bitplane) with subsequent image analysis performed using either Fiji (Schindelin et al., 2012) or Imaris (Bitplane). Images were manually quantified by circling visible nuclei within a single z-plane to generate regions of interest (ROIs) using the ROI manager tool in Fiji. The average grey values were then measured for the selected regions for each fluorescent channel.

### Light-sheet microscopy

The unfixed trunk area of E9.5 $Wt1^{CreERT2/+}$;$Rosa26^{mTmG/+}$ embryos dosed with Tam at E8.5 was mounted in 1% low melting point agarose (ROTI Garose, Carl Roth), transferred into a capillary (Size 4, Brand GmbH) and placed in the imaging chamber of a Zeiss light-sheet Z.1 microscope with one 5× detection objective and two 5× illumination objectives supported with Zen software (Carl Zeiss). Imaging was performed using the 488 nm channel to capture GFP-positive cells, and the 561 nm channel to image membrane-bound dTomato to provide tissue context. Images were adjusted to colours using Imaris software and the movie edited to highlight the shape of the selected area by using the surface tool, and by labelling GFP-positive cells with coloured spots to indicate their different locations.

### Analysis of single-cell mouse gastrulation atlas

Data analysis from a previously published dataset (Pijuan-Sala et al., 2019) was performed using R (version 4.3.2). Data were accessed through Bioconductor 'MouseGastrulationData' package (version 1.16.0), and only processed data were extracted for downstream analysis. Extracted data were normalised using the 'logNormCounts' function from the 'scuttle' package (version 1.12.0). Normalisation of cell count per sample was carried out by dividing the total number of cells by the sample size and rounding up to the nearest integer. Signature scoring was performed using the AUCell package (version 1.26.0) (Aibar et al., 2017) and threshold was manually selected from suggested thresholds calculated by the package. All plots were created using the 'ggplot2' package (version 3.4.3).

### Analysis of whole-mount embryos

For whole-mount CD31 staining analysis, regions of interest (indicated by white boxes) were pre-processed by background subtraction, median filtering and manual thresholding due to photobleaching. Each ROI was then: (1) inputted into 3Dsuite to measure the total volume of the vasculature of respective regions; and (2) 3D-skeletonised and the resulting branching information was analysed using AnalyzeSkeleton.

### Acknowledgements

We acknowledge funding by the BBSRC Alert13, grant number BB/L014947/1, for the Zeiss Z.1 light-sheet microscope. We are also indebted to the University of Liverpool's Centre for Cell Imaging (CCI) facility for provision of state-of-the-art imaging equipment funded through grants awarded to the University of Liverpool's CCI by the MRC (MR/K015931/1) and the BBSRC (BB/M012441/1, BB/R01390X/1), and for excellent technical support, assistance and training. We express our thanks to the Biomedical Services Unit at the University of Liverpool, for expert support in mouse maintenance, breeding and timed mating. We acknowledge Sumaya Dauleh and Thomas P. Wilm for technical assistance.

### Competing interests

The authors declare no competing or financial interests.

### Author contributions

Conceptualization: B.W.; Data curation: D.A.T., B.W.; Formal analysis: S.H.A., H.T.N., S.B., D.A.T., B.W.; Funding acquisition: S.B., D.A.T., B.W.; Investigation: S.H.A., H.T.N., L.L., S.L., S.B., L.C., M.B., D.A.T., B.W.; Methodology: S.H.A., H.T.N., L.L., S.L., S.B., L.C., M.B., D.A.T., B.W.; Project administration: D.A.T., B.W.; Resources: D.A.T., B.W.; Supervision: D.A.T., B.W.; Validation: D.A.T., B.W.; Visualization: S.H.A., H.T.N., L.L., D.A.T., B.W.; Writing – original draft: H.T.N., D.A.T., B.W.; Writing – review & editing: S.H.A., H.T.N., L.L., S.L., S.B., D.A.T., B.W.

### Funding

B.W. was supported by a Medical Research Council project grant (MR/M012751/1) and the Biotechnology and Biological Sciences Research Council (BBSRC) Doctoral Training Partnership scheme (BB/M011186/1 to M.B.; BB/T008695/1 to H.T.N.), and acknowledges generous funding from the Al-Baha University, Saudi Arabia (PhD Studentship to S.H.A.). S.B. was funded by a National Health Service Student Bursary. L.L. and L.C. were MRes students self-funding their studies. D.A.T. was supported in this work by a BBSRC New Investigator Grant (BB/X000907/1 to S.L.) and a David Sainsbury NC3Rs (National Centre for the Replacement, Refinement and Reduction of Animals in Research) Fellowship (NC/P001467/1). Open Access funding provided by the University of Liverpool. Deposited in PMC for immediate release.

### Data and resource availability

Data and code are available on https://github.com/hueiteng99/wt1_smc_2025.git.

### The people behind the papers

This article has an associated 'The people behind the papers' interview with some of the authors.

### Peer review history

The peer review history is available online at https://journals.biologists.com/dev/lookup/doi/10.1242/dev.204332.reviewer-comments.pdf

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
