## [Peer Review File · Development (Cambridge, England)]

Dynamic WT1 expression during gastrulation specifies peritoneal smooth muscle fate independently from mesothelial fate

Suad Hassan Alsukari, Huei Teng Ng, Lilly Lang, Sharna Lunn, Shanthi Beglinger, Lauren Carr, Michael Boyes, David Andrew Turner and Bettina Wilm

DOI: 10.1242/dev.204332

Editor: Liz Robertson

Review timeline

Original submission:	19 August 2024
Editorial decision:	1 October 2024
First revision received:	2 May 2025
Editorial decision:	2 June 2025
Second revision received:	6 June 2025
Accepted:	8 June 2025

Original submission

First decision letter

MS ID#: dev.204332

MS TITLE: Dynamic Wt1 expression in the gastrulation-stage mouse embryo specifies vascular and visceral smooth muscle cell fate independently from mesothelial fate

AUTHORS: Suad Hassan Alsukari, Huei Teng Ng, Lilly Lang, Sharna Lunn, Shanthi Beglinger, Lauren Carr, Michael Boyes, David Andrew Turner and Bettina Wilm

Dear Dr Wilm,

I have now received all the referees' reports on the above manuscript, and have reached a decision. The referees' comments are appended below, or you can access them online: please go to:

As you will see, the referees express considerable interest in your work, but have some significant criticisms and recommend a substantial revision of your manuscript before we can consider publication. If you are able to revise the manuscript along the lines suggested, which may involve further experiments, I will be happy receive a revised version of the manuscript. Your revised paper will be re-reviewed by one or more of the original referees, and acceptance of your manuscript will depend on your addressing satisfactorily the reviewers' major concerns. Please also note that Development will normally permit only one round of major revision. If it would be helpful, you are welcome to contact us to discuss your revision in greater detail. Please send us a point-by-point response indicating your plans for addressing the referees' comments, and we will look over this and provide further guidance.

Please attend to all of the reviewers' comments and ensure that you clearly highlight all changes made in the revised manuscript. Please avoid using 'Tracked changes' in Word files as these are lost in PDF conversion. I should be grateful if you would also provide a point-by-point response detailing how you have dealt with the points raised by the reviewers in the 'Response to Reviewers' box. If you do not agree with any of their criticisms or suggestions please explain clearly why this is so.

Reviewer 1*Advance summary and potential significance to field*

Alsukari, Wilm and colleagues investigated the role of the Wilms' Tumour protein (Wt1) in the development of the mesothelium and visceral smooth muscle cells (visSMCs) and vascular smooth muscle cells (vSMCs), especially around the mouse intestine. In contrary to current literature, the authors suggest that these intestinal smooth muscles lineages don't arise from the mesothelium (coelomic epithelium), which has been up-to-date described as the order of events. Instead, the authors suggest that Wt1-expression during gastrulation (E7.5-E8.5) and within the early lateral plate mesoderm (LPM) delineates a distinct population of smooth muscle precursor cells. Using tamoxifen-induced lineage tracing, they showed that Wt1-expressing cells can give rise to visSMCs and vSMCs already before E9.5 but not really mesothelium, while those cells contribute later (E9.5 onwards) and independently to the mesothelial lineage. Re-analysis of a public scRNA-seq dataset (Pijuan-Sala et al. 2019) and immunofluorescence imaging described Wt1 expression in epiblast, primitive streak, and emerging mesodermal progenitors, earlier than previously documented in mouse, and showed co-expression with several key smooth muscle markers. The study concludes that mesothelium and smooth muscle lineages develop independently from Wt1-expressing cells at different stages.

The authors uncovered a potential role of Wt1 in the temporal differentiation of two LPM-derived lineages: the smooth muscles (visceral and vasculature) and the mesothelium. The role of Wt1 in vSMCs has been described in multiple papers (including those of the authors), but a potential dual role of Wt1 in smooth muscle and mesothelium differentiation had not been considered before. While it is well-described that mesothelium and smooth muscles derive from the LPM, how those lineages do so is still quite unstudied. This rather descriptive study adds some insights about how Wt1, among other genes, during the differentiation of the mesothelium and the smooth muscles, which can also highly dependent on the organ system the mesothelium and SMCs become part of. I believe this will be of interest for those interested in the mesothelium and smooth muscle development, and lays the foundation for others to explore gene regulation and cell fate specification these understudied tissue more thoroughly. This also raises questions about how these insights could shed light on developmental defects such as congenital hernias (e.g., omphalocele, gastroschisis), though such implications were not explored or highlighted in detail by the authors.

Comments for the author

The authors present a range of imaging datasets, including genetic lineage tracing and immunofluorescence, along with the use of a publicly available scRNA-seq dataset on early mouse development. I suggest to highlight better throughout the text that this emphasis is on the observed role of Wt1 pertains specifically to the intestine (mid-gut) and cannot yet be extended to the fore-gut or hind-gut, where contributions to vascular SMCs are absent (Sup. Table 1). (Also mention in Title).

While the data is interesting, I believe it does not fully support the conclusions. Major revisions are recommended, including:

- The scRNA-seq data, though used effectively, could benefit from improved representation and interpretation. I provide specific suggestions below:

- First, I suggest visualizing Wt1 expression on a UMAP projection at key time points, along with the cluster annotations from the Puijan-Sala dataset. This would support the bar plots in Fig. 3A, B, and Sup. Fig. 3.1C, D. Wt1-expressing cells are sparse in E6.5-E7.5 embryos, and the overlap between Wt1 and Bra (Fig. 3B) is not convincing, particularly when viewed in the public Shiny app (<https://marionilab.cruk.cam.ac.uk/MouseGastrulation2018/>), where at E7.5, there is no clear overlap between Wt1 and Bra (T) in UMAP. How do the authors explain this?

- The authors should discuss dropout effects in scRNA-seq data, as transcription factors can often be under-detected, leading to false zeros. Over-interpretation of expression percentages or normalized counts should be avoided. The mean expression values used throughout the text are not so informative, especially when correlating counts between transcription factors and genes

encoding structural proteins or receptors (Tagln, Pdgfrb). This analysis is quite flawed and should be reconsidered.

- While many plots and analyses are presented in Fig. 3, the key messages are unclear. The "trajectories" lack clarity, as they don't show co-expression of key markers. The dot plots in Sup. Fig. 3.1E, F are more insightful (but consider adding Wt1) and may be more useful than some bar plots in Fig. 3. Co-expression between genes could be better visualized using correlation plots. For instance, a "smooth muscle score" (signature) could be calculated and compared against Wt1 expression in the mesenchymal/LPM clusters from E7.5-E9.5. A similar "mesothelium score" could be generated. Potentially, even the mesothelium and the smooth muscle score could be co-plotted.

- The imaging results show that Wt1 contributes to visceral and vascular SMCs primarily around the intestine. These insights could be better linked to the single cell data. Since the single cell data set includes the whole embryo, so also non-Wt1-derived smooth muscle and mesothelium populations.

- To support the hypothesis that Wt1 defines two distinct populations post-gastrulation, it would be interesting to examine if there are differential gene regulatory networks in smooth muscle and mesothelial progenitors. How do the authors view Hand1/2's role relative to Wt1 in these lineages, as Hand1/2 are highly expressed at relevant time points and are documented as early regulators in both mesothelial and smooth muscle progenitors?

- The Wt1 lineage tracing appears patchy in most induction (pulse) and endpoint (chase) images, making interpretation difficult. Can the authors clarify if this is due to technical factors like tamoxifen treatment or biological variation?

- The study remains descriptive, lacking an experimental manipulation showing that smooth muscle development is independent of mesothelium. A Wt1 loss-of-function experiment could clarify this. The authors could use a Wt1 ablation model (e.g. they used in Wilm et al., 2021). A comparison with Hand1/2 studies, where knock-down/out affects SMC migration and mesothelial defects, would strengthen the findings. Ablation of Wt1 at E7.5-E8.5 could validate whether SMC progenitors contribute to smooth muscles independently, while ablation from E9.5 onward would affect both lineages.

- Given Hand1/2's importance in vis/vSMC and mesothelial development, co-expression of Wt1 and these proteins should be shown in Figures 4 and/or 5. Good antibodies for mouse immunofluorescence are available.

To help improve the readability of the manuscript, I have the following minor suggestions:

- * About the figures in general: make some cartoons / schematic representations about the anatomy of the studied structures in the early mouse embryos, to help those of us unfamiliar with the structures.

- * Figure 1: add end-stage timepoints for each figure panel.

- * The Rosa26mTmG can already be in the Results text introduced as a fluorescent reporter line.

- * Abbreviations are not consistently used throughout the text (scrRNA-seq, visSMC, etc.) and not all introduced from the first mention.

- * Related to Sup. Fig. 2 and the Results text: the spleen mentioning is a bit odd. Both in the figure panels, the spleen is not mentioned / indicated at any time point. This could be better described and reasoned why mentioned in the first place.

- * Figure 2: I would suggest the authors to first show Tamoxifen @ E8.5 and then @ E11.5, following the text. Moreover, arrows from E are missing in E'. Additionally, the magenta cells are hard to spot in the images of E and F. Interesting areas could benefit from an additional zoom in / representation in grey scale. Lastly, the outline of the embryo / tissue in each section could be highlighted by dashed white lines.

- * Figure 4: add end-stage timepoints for each embryo in the panel. Also in this figure, especially for panels I-N, the single channels could be all represented in grey scale to improve visualization.

- * Figure 5: misses a Tamoxifen treatment schematic. It is unclear if the staining in the fore- and hind limb buds in C are background or actual labeling due to Wt1-lineage tracing. The panels A, B are not mentioned specifically in the text.
- * References for classical mesoderm and LPM markers are missing throughout the text.
- * Figure 6: in the Results text is missing that Tamoxifen was administered @ E8.5.

Reviewer 2

Advance summary and potential significance to field

In the study by Alsukari et al., the authors use a Wt1CreERT2 mouse model with either the Rosa26LacZ or Rosa26mTmG reporter systems to demonstrate the temporal and spatial relationship between emerging smooth muscle cells and the expression of WT1 during gastrulation and post-gastrulation stages of mouse embryonic development. Their analysis also reveals that WT1 is expressed in a pool of progenitor cells within the emerging mesoderm of the gastrulating embryo. However, despite these interesting and novel findings, the role of WT1 in these progenitor cells was not investigated.

Comments for the author

1. The Wt1CreERT2 mouse model used in this study is a knock-in (KI) at exon 1. Since this is the first study to report such early expression of WT1, I suggest considering a modification in the order of data presentation. It seems more logical to first examine WT1 expression early in development and then conduct lineage tracing experiments by inducing recombination at these earlier stages.
2. WT1 is required for the formation of several tissues and organs during embryonic development. What is the role of WT1 in this pool of early progenitor cells? Is WT1 expression in these progenitors essential for the formation of vSMCs and visSMCs? The authors have previously used the Wt1CreERT2/Loxp model combined with the Rosa26LacZ reporter to investigate its role in maintaining the visceral serosa and intestinal wall (Wilm TP, 2021). A similar approach, in combination with the mTmG reporter as shown in Figures 5 and 6, could be used to determine whether WT1 deletion in these early progenitor cells affects the formation of their derivatives, including vSMCs and visSMCs.
3. Have the authors considered administering tamoxifen before E7.5? As noted in line 402, a previous study using a constitutive Wt1Cre KI mouse model identified GFP-expressing cells in the developing heart and throughout the embryo starting from E8.5 (Rudat C, 2012). Given the data presented in this manuscript, which demonstrate very early WT1 expression, and considering the temporal delay of approximately 0.5 embryonic days between tamoxifen administration and reporter expression, do the authors think that administering tamoxifen at an earlier stage might help broaden the understanding of the contribution of this early population of WT1-progenitor cells?

Minor issues:

- * Gene nomenclature should follow established conventions. Please note that the name of the Wt1 gene has been updated.
Mouse: <http://www.informatics.jax.org/mgihome/nomen/index.shtml>
- * To improve comprehension, the labelling of some figures should be modified. In Figure 1, please use the same labelling format as in Figure 2, including the embryonic day of tamoxifen administration and the day of analysis. Additionally, for Figure 4, please include the embryonic stages.
- * Figures 1 and Supplementary Figures 1 and 2 show whole-mount staining data. While the authors claim to identify specific cell types based on each tamoxifen scheme, a whole-mount approach alone is insufficient for determining the precise type of cells that are LacZ+ or GFP+ (lines 136-151). The conclusions drawn from the data presented in these figures should be revised to

accurately reflect the information they provide. Figure 2, where the authors have conducted a double immunostaining analysis, provides a more detailed characterization of the GFP+ cells.

Reviewer 3

Advance summary and potential significance to field

The manuscript by Alsukari and colleagues examines the temporal relationship between the mesothelial and vascular smooth muscle cell lineages. They clearly demonstrate that intestinal vascular and visceral smooth muscle cells develop independently from the visceral mesothelium, redefining the mechanism for their formation.

Comments for the author

The data presented both in the main text and supplementary figures clearly supports their conclusion of an early emergence of visSMC and vSMC lineage in the gastrulating embryo, and an uncoupling from the mesothelial lineage.

There are no major concerns with the experimental design, data presented or the conclusions drawn. I think the manuscript would be improved by including Supplementary fig 9 in the main body of the manuscript to help summarise the conclusions drawn from this study. There are a lot of supplementary figures in the manuscript which although descriptive, are important to help draw the conclusions outlined in the manuscript.

First revision

Author response to reviewers' comments

Response to Reviewers' comments:

In response to the reviewers' comments, we have reorganised parts of the manuscript by adding data that we obtained by performing additional experiments, and by moving figures around. This is reflected by changes to large parts of the manuscript, which we have highlighted in the accompanying revised manuscript (version with highlights). We believe that these changes have made the manuscript stronger and clearer and underlined our overall message. We are therefore grateful for the constructive comments the reviewers provided to improve our work.

Reviewer 1:

We thank the reviewer for the many insightful comments and suggestions to our manuscript. The reviewer writes:

1. *In contrary to current literature, the authors suggest that these intestinal smooth muscles lineages don't arise from the mesothelium (coelomic epithelium), which has been up-to-date described as the order of events. The role of Wt1 in vSMCs has been described in multiple papers (including those of the authors), but a potential dual role of Wt1 in smooth muscle and mesothelium differentiation had not been considered before.*

We thank the reviewer for this important point. The hypothesis that the intestinal smooth muscle arise from the mesothelium was originally formulated by our work published in Wilm et al., 2005. Most of the results from this and related studies had been obtained using a *WT1-Cre* mouse line that is a transgene where Cre recombinase is expressed under control of a human *WT1* promoter (Que et al., 2008; Wilm et al., 2005; Wilm and Munoz-Chapuli, 2016). This approach does not allow a temporal dissection of lineage events. The lineage tracing therefore was not as sensitive as with a temporally-controlled Cre recombinase. Other lineage tracing experiments using non-inducible *Wt1-Cre* mouse lines

also showed a contribution of *Wt1*-expressing cells in the vascular and visceral smooth muscle (see (Carmona et al., 2013)).

As discussed in the manuscript on lines 74-85, all these data obtained from non-inducible *Wt1*-driven Cre mouse lines are therefore in agreement.

However, experiments using all these mouse lines suffered from the lack of temporal control. Only the temporal control of recombination activation can convey *when* during embryonic development the cells are labelled, which then can provide information of their lineage origin based on the information regarding tissue identity and developmental origins, available at the time. Therefore, the present is the first study to provide this insight, demonstrating that two separate lineages of *Wt1*-expressing cells arise with different signatures.

2. *I suggest to highlight better throughout the text that this emphasis is on the observed role of Wt1 pertains specifically to the intestine (mid-gut) and cannot yet be extended to the fore-gut or hind-gut, where contributions to vascular SMCs are absent (Sup. Table 1). (Also mention in Title).*

We thank the reviewer for this point. We have provided evidence of visceral smooth muscle contribution of *Wt1*-derived lineage traced cells in the small intestine, the stomach and in the hindgut (see Supplementary Figure 2). In order to provide focus of the study, we have adjusted the title to reflect that the tissues analysed are linked to the peritoneum.

3. *First, I suggest visualizing Wt1 expression on a UMAP projection at key time points, along with the cluster annotations from the Puijan-Sala dataset. This would support the bar plots in Fig. 3A, B, and Sup. Fig. 3.1C, D.*

Thank you for this suggestion which we have carefully considered. We direct the reviewer to the fact that the UMAP analysis in the public Shiny app fails to provide the resolution that our analysis has generated. We have adjusted the original graphs in Figure 3 and Supplementary Figures to show

- all *Wt1*-expressing cells in all tissues throughout stages E6.5, E7.0, E7.25, E7.5, E7.75, E8.0, E8.25 and E8.5 (Supplementary Figure 3.1C);
- *Wt1*-expressing cells in all tissues that co-expressed either at least 3 genes of the SMC signature or at least 3 genes of the mesothelial cell signature (Figure 3D, E).

We have included, below, screenshots from the Shiny app to illustrate the presence and localisation of *Wt1*-expressing cells within the different tissues. This information is much less informative than the data we have included in the revised Figures.

4. *Wt1-expressing cells are sparse in E6.5-E7.5 embryos, and the overlap between Wt1 and Bra (Fig. 3B) is not convincing, particularly when viewed in the public Shiny app*

(<https://marionilab.cruk.cam.ac.uk/MouseGastrulation2018/>) [Pijuan-Sala, B., Griffiths, J. A., Guibentif, C., Hiscock, T.W., Jawaid, W., Calero-Nieto, F. J., Mulas, C., Ibarra-Soria, X., Tyser, R. C. V., Ho, D. L. L. et al. (2019). A single-cell molecular map of mouse gastrulation and early organogenesis. *Nature* **566**, 490-495. doi:10.1038/s41586-019-0933-9], where at E7.5, there is no clear overlap between *Wt1* and *Bra* (*T*) in UMAP. How do the authors explain this?

Co-expression between genes could be better visualized using correlation plots.

Thank you for this comment. As discussed under point 3, we would like to direct the reviewer towards the fact that the UMAP analysis in the public Shiny app doesn't provide the resolution that our analysis has generated. We have provided Supplementary Figure 3.1D where all *Wt1*- and *Bra*- co-expressing cells are displayed according to the tissue annotation of the Shiny app. We have also displayed correlation between *Wt1* and *Brachyury* expression in Figure 3B. While there are only very few cells co-expressing the transcripts for both *Wt1* and *Bra* from E6.5 onwards, at E7.0 and E7.5 there is statistical significance in the correlation of expression between the two markers.

5. The authors should discuss dropout effects in scRNA-seq data, as transcription factors can often be under-detected, leading to false zeros. Over-interpretation of expression percentages or normalized counts should be avoided. The mean expression values used throughout the text are not so informative, especially when correlating counts between transcription factors and genes encoding structural proteins or receptors (*Tagln*, *Pdgfrb*). This analysis is quite flawed and should be reconsidered.

The reviewer raises a few important points regarding scRNAseq data and their analysis:

- A) There is a risk of dropout effects where transcription factors are often under-detected because of their low expression levels, leading to false zeros. We have tried to address this by showing levels of expression above zero.
- B) We have included normalised cell counts because the number of embryos analysed by Pijuan-Sala and colleagues is not consistent between different embryonic stages (see Supplementary Figure 3.1A). In stages where more embryos have been analysed, more cells would therefore be included in the datasets without

normalisation. We consider normalised cell counts the best way to handle this variation and therefore display the mean cell counts (see Supplementary Figure 3.1C, D, Figure 3D, E, Supplementary Figure 3.2E)

6. *While many plots and analyses are presented in Fig. 3, the key messages are unclear. The "trajectories" lack clarity, as they don't show co-expression of key markers. For instance, a "smooth muscle score" (signature) could be calculated and compared against Wt1 expression in the mesenchymal/LPM clusters from E7.5-E9.5. A similar "mesothelium score" could be generated. Potentially, even the mesothelium and the smooth muscle score could be co-plotted.*

We thank the reviewer for this excellent suggestion. We have used Muhl et al., 2022 (<https://doi.org/10.1016/j.devcel.2022.09.015>) to generate a set of genes we consider a smooth muscle signature, and Kanamori-Katayama et al., 2011 (<https://doi.org/10.1371/journal.pone.0025391>) to generate a set of genes that we consider a mesothelium signature (see Supplementary Figure 3.1G). Analysis using these two signatures against the scRNAseq data in the context of *Wt1* expression, revealed two main findings:

1. *Wt1*-expressing cells which co-expressed the SMC gene set (signature), are more abundant at E6.5 and E8.5 than *Wt1*-expressing cells co-expressing the mesothelial signature, and there are higher expression levels in the SMC signature-defined cells (Figure 3A).
2. Correlation graphs of SMC or MC signatures in *Wt1*-expressing cells showed that more genes of the SMC signatures start to be highly significantly expressed at earlier stages than genes of the mesothelial signatures (Figure 3B). Plotting the *Wt1*-expressing cells with SMC vs mesothelial cell signature at E7.5 reveals two separate populations with 11 (SMC) and 10 (mesothelial) cells, respectively (Figure 3F, H). At E8.5, the two groups have increased in size, interestingly with an overlap of 11 cells (Figure 3G, I).
3. Expression levels of *Wt1* are on average lower in the SMC signature group than in the mesothelial cell signature group (Figure 3H, I).

These data suggest that there is already an earlier separation into SMC and mesothelial lineage, starting from the E6.5 onwards.

The dot plots in Sup. Fig. 3.1E, F are more insightful (but consider adding Wt1) and may be more useful than some bar plots in Fig. 3.

We thank the reviewer for this point. We had included the dot plot to demonstrate that the term 'mesenchyme' used within the Pijuan-Sala et al., 2019 data sets, describes in fact lateral plate mesoderm. We have re-organised this dot plot, and included *Wt1* expression. We have also added dot plots for the signature of mesothelial and smooth muscle genes in Figure 3.

Taking together suggestions from points 3-6, we have redesigned Figure 3 and Supplementary Figures 3.1 and 3.2.

7. *The imaging results show that Wt1 contributes to visceral and vascular SMCs primarily around the intestine. These insights could be better linked to the single cell data.*

Thank you very much for this suggestion. We have made clear that visceral and vascular smooth muscle cells that are found surrounding the intestine, are lateral plate mesoderm-derived (p. 4, lines 103-108). A detailed understanding of the molecular processes controlling lineage differentiation of the visceral and vascular smooth muscle cells surrounding the intestinal tract has not been provided yet. Therefore, our data are the first to generate this clear link back to the emerging lateral plate mesoderm.

8. *To support the hypothesis that Wt1 defines two distinct populations post-gastrulation, it would be interesting to examine if there are differential gene regulatory*

networks in smooth muscle and mesothelial progenitors.

Thank you very much for this suggestion. We isolated a group of *Wt1*-expressing cell with co-expression of at least 3 or more SMC or MC markers (signature) in E8.5 (Figure 3G) and attempted to perform pseudobulk differential analysis on the 58 SMCs and 33 mesothelial cells. However, due to the small sample size, it was not possible to extract meaningful results based on the differential analysis and hence we weren't able to carry forward a gene regulatory network analysis.

9. *How do the authors view Hand1/2's role relative to Wt1 in these lineages, as Hand1/2 are highly expressed at relevant time points and are documented as early regulators in both mesothelial and smooth muscle progenitors?*

We thank the reviewer for this insightful comment. We included correlation graphs between the SMC and MC signatures and *Hand1* and *Hand2* transcripts in Supplementary Figure 3.2A, B, as well as correlation graphs between the two signatures and *Hand2* transcripts in all *Wt1*-expressing cells (Supplementary Figure 3.2C) and *Hand2* transcripts in all *Wt1*-expressing mesoderm cells (Supplementary Figure 3.2D). We also show the normalised number of *Wt1*- and *Hand2*-co-expressing cells in Supplementary Figure 3.2E.

We have described and commented on these findings on pages 8-9, lines 221-241. We discussed *Hand1* and *Hand2* and their potential roles in comparison to *Wt1* in the context of our findings, in lines 462-487.

10. *The Wt1 lineage tracing appears patchy in most induction (pulse) and endpoint (chase) images, making interpretation difficult. Can the authors clarify if this is due to technical factors like tamoxifen treatment or biological variation?*

We observed indeed patchy distribution of lineage-traced cells, especially clearly visible in the mesothelial covering. We previously made this observation in postnatal lineage tracing with this system - see images in Wilm et al., 2021 (<https://doi.org/10.1038/s41598-021-95380-1>) . We interpret this as an inherent result of the *Wt1^{CreERT2}* lineage tracing system. We have observed it independently of route of tamoxifen administration, and added a section in the discussion to raise awareness of this fact (lines 418-425).

11. *The study remains descriptive, lacking an experimental manipulation showing that smooth muscle development is independent of mesothelium. A Wt1 loss-of-function experiment could clarify this. The authors could use a Wt1 ablation model (e.g. they used in Wilm et al., 2021). Ablation of Wt1 at E7.5-E8.5 could validate whether SMC progenitors contribute to smooth muscles independently, while ablation from E9.5 onward would affect both lineages. A comparison with Hand1/2 studies, where knock-down/out affects SMC migration and mesothelial defects, would strengthen the findings.*

We thank the reviewer for this suggestion. We have performed ablation of *Wt1* at E7.5 and analysed embryos at E12.5; this is described in the new Figure 6. Our results show that there is clearly an effect of loss of *Wt1* on the developing vasculature in the intestine and mesentery, with much higher variability in endothelial cell density and endothelial network formation in the mutant than in the control embryos, suggesting irregular vasculature formation. Furthermore, we observed that vascular SMCs were less well aligned with the developing endothelial vessels.

These results clearly demonstrate a functional relationship between *Wt1* expression in gastrulation-stage embryos, and later vascular development in the intestinal tract.

12. *Given Hand1/2's importance in vis/vSMC and mesothelial development, co-expression of Wt1 and these proteins should be shown in Figures 4 and/or 5. Good antibodies for mouse immunofluorescence are available.*

We thank the reviewer for the suggestion. We have performed additional staining for *Hand2*

with Wt1 given its relevance uncovered in the analysis of the scRNAseq data, as shown in Supplementary Figure 3.3. As addressed already under point 9, we have also done a correlation analysis between Wt1 and Hand1 or Hand2 and also a deeper analysis with respect to the SMC and MC signature markers.

The data suggest that there are clearly similarities in the contribution of Hand2 and Wt1 to SMC and mesothelial fate and in their expression patterns, since we observed co-expression in several cells in the posterior LPM in E8.5 mouse embryos.

13. *Make some cartoons / schematic representations about the anatomy of the studied structures in the early mouse embryos, to help those of us unfamiliar with the structures.*

We had provided schematics for illustration in Figure 4D and Supplementary Figure 4 for E8.5 posterior mouse embryo. We have now added a schematic for Figure 2, as requested (see point 18), to provide additional clarity.

14. *Figure 1: add end-stage timepoints for each figure panel.*

We thank the reviewer for the suggestion to improve clarity and have added the end-stage time points to the Figure.

15. *The Rosa26mTmG can already be in the Results text introduced as a fluorescent reporter line.*

We thank the reviewer for raising this point which we have addressed in line 125.

16. *Abbreviations are not consistently used throughout the text (scRNA-seq, visSMC, etc.) and not all introduced from the first mention.*

We thank the reviewer for raising this point which we have addressed throughout.

17. *Related to Sup. Fig. 2 and the Results text: the spleen mentioning is a bit odd. Both in the figure panels, the spleen is not mentioned / indicated at any time point. This could be better described and reasoned why mentioned in the first place.*

We have adjusted the figure legends for Supplementary Figure 1 to refer more directly to the staining in the spleen. We have also placed the results about the spleen into context, see lines 147-151.

18. *Figure 2: I would suggest the authors to first show Tamoxifen @ E8.5 and then @ E11.5, following the text. Moreover, arrows from E are missing in E'. Additionally, the magenta cells are hard to spot in the images of E and F. Interesting areas could benefit from an additional zoom in / representation in grey scale. Lastly, the outline of the embryo / tissue in each section could be highlighted by dashed white lines.*

We thank the reviewer for raising these points for added clarity, which we have addressed, see Figure 2. In addition, we have added a schematic to highlight the areas shown within the sections.

19. *Figure 4: add end-stage timepoints for each embryo in the panel. Also in this figure, especially for panels I-N, the single channels could be all represented in grey scale to improve visualization.*

We thank the reviewer for this point. We would like to politely point out that out that no lineage tracing is shown in this figure, but instead the E8.5 mouse posterior embryo is shown in high resolution, including the primitive streak, emerging notochord, LPM/IM, neuromesodermal progenitor cells, node, endoderm, allantois and primitive blood cells (Figure 4D).

We have added annotations for Figure 4I-N, and in addition, have provide grey scale images of the individual fluorescence for each marker.

20. *Figure 5: misses a Tamoxifen treatment schematic. It is unclear if the staining in the fore- and hind limb buds in C are background or actual labeling due to Wt1-lineage tracing. The panels A, B are not mentioned specifically in the text.*

We thank the reviewer for this point to add additional clarity. We have provided a treatment schematic to this figure which we have re-organised into Supplementary Figure 6.2; we have added one also to Supplementary Figure 6.1. We have added a sentence to the figure legend regarding the brightly appearing limb buds in this Supplementary Figure 6.2. We have added mentions of panels A, B in the text, see line 290.

21. *References for classical mesoderm and LPM markers are missing throughout the text.*

We have added relevant references to the text.

22. *Figure 6: in the Results text is missing that Tamoxifen was administered @ E8.5.*

We thank the reviewer for this point and have added this to the text, lines 287-289.

Reviewer 2: SUMMARY OF THE ADVANCE MADE IN THIS PAPER AND ITS POTENTIAL SIGNIFICANCE TO THE FIELD

1. *The Wt1CreERT2 mouse model used in this study is a knock-in (KI) at exon 1. Since this is the first study to report such early expression of WT1, I suggest considering a modification in the order of data presentation. It seems more logical to first examine WT1 expression early in development and then conduct lineage tracing experiments by inducing recombination at these earlier stages.*

We thank the reviewer for this comment. We would like to point out that the rationale for this manuscript was the quest to identify the population of *Wt1*-expressing cells in the embryo that would later contribute to the vascular and visceral smooth muscle. This was to further clarify the temporal processes that link *Wt1* expression to smooth muscle development.

We feel the order of how the analysis is being described, is more in line with this aim and provides the rationale for performing the study in the first place. We would therefore like to decline the suggestion to change the order of the narrative.

2. *WT1 is required for the formation of several tissues and organs during embryonic development. What is the role of WT1 in this pool of early progenitor cells? Is WT1 expression in these progenitors essential for the formation of vSMCs and visSMCs? The authors have previously used the Wt1CreERT2/Loxp model combined with the Rosa26LacZ reporter to investigate its role in maintaining the visceral serosa and intestinal wall (Wilm TP, 2021). A similar approach, in combination with the mTmG reporter as shown in Figures 5 and 6, could be used to determine whether WT1 deletion in these early progenitor cells affects the formation of their derivatives, including vSMCs and visSMCs.*

We thank the reviewer for this suggestion, which is similar to the one raised by reviewer 1. As addressed above in point 11/12 (reviewer 1), we performed ablation of *Wt1* at E7.5 and analysed embryos at E12.5 (new Figure 6). We found that loss of *Wt1* at this stage results in changes in the developing vasculature in the intestine and mesentery in mutant embryos as highlighted much higher variability in endothelial cell density and endothelial network formation when compared to control embryos. We also analysed SMC development using SMA (ACTA2) and found that vascular SMCs were less well aligned with the developing vessels.

These results clearly demonstrate a functional relationship between Wt1 expression in gastrulation-stage embryos, and later vascular development in the intestinal tract.

3. *Have the authors considered administering tamoxifen before E7.5? As noted in line 402, a previous study using a constitutive Wt1Cre KI mouse model identified GFP-expressing cells in the developing heart and throughout the embryo starting from E8.5 (Rudat C, 2012). Given the data presented in this manuscript, which demonstrate very early WT1 expression, and considering the temporal delay of approximately 0.5 embryonic days between tamoxifen administration and reporter expression, do the authors think that administering tamoxifen at an earlier stage might help broaden the understanding of the contribution of this early population of WT1-progenitor cells?*

We thank the reviewer for raising this point. We struggled to get successful matings for several months, but finally managed to obtain 3 litters. Unfortunately, we only observed in one embryo very few labelled cells, near the gonad, the intestine and in the liver, when giving Tamoxifen at E6.5, and analysing at E14.5. This is shown in the new Supplementary Figure 6.3.

4. *Gene nomenclature should follow established conventions. Please note that the name of the Wt1 gene has been updated.*

Mouse: <http://www.informatics.jax.org/mgihome/nomen/index.shtml>

The reviewer has correctly pointed out that the nomenclature for mouse genes and proteins is as follows: *Wt1* as gene name, WT1 as protein name:

<https://www.informatics.jax.org/mgihome/nomen/index.shtml>

We have applied this throughout the text and also to the other marker genes/proteins.

5. *To improve comprehension, the labelling of some figures should be modified. In Figure 1, please use the same labelling format as in Figure 2, including the embryonic day of tamoxifen administration and the day of analysis. Additionally, for Figure 4, please include the embryonic stages.*

We thank the reviewer for this suggestion. We have provided additional information in Figures 2, 4 and 5 to provide clarity of stages analysed.

6. *Figures 1 and Supplementary Figures 1 and 2 show whole-mount staining data. While the authors claim to identify specific cell types based on each tamoxifen scheme, a whole-mount approach alone is insufficient for determining the precise type of cells that are LacZ+ or GFP+ (lines 136-151). The conclusions drawn from the data presented in these figures should be revised to accurately reflect the information they provide. Figure 2, where the authors have conducted a double immunostaining analysis, provides a more detailed characterization of the GFP+ cells.*

Reviewer 3:

I think the manuscript would be improved by including Supplementary fig 9 in the main body of the manuscript to help summarise the conclusions drawn from this study.

We thank the reviewer for this suggestion. We have added this as Figure 7 to the main body of the manuscript.

References

- Carmona, R., Cano, E., Mattiotti, A., Gaztambide, J. and Munoz-Chapuli, R. (2013). Cells derived from the coelomic epithelium contribute to multiple gastrointestinal tissues in mouse embryos. *PLoS one* **8**, e55890.
- Que, J., Wilm, B., Hasegawa, H., Wang, F., Bader, D. and Hogan, B. L. (2008). Mesothelium contributes to vascular smooth muscle and mesenchyme during

lung development. *Proc Natl Acad Sci U S A* **105**, 16626-16630.

Wilm, B., Ipenberg, A., Hastie, N. D., Burch, J. B. E. and Bader, D. M. (2005). The serosal mesothelium is a major source of smooth muscle cells of the gut vasculature. *Development* **132**, 5317-5328.

Wilm, B. and Munoz-Chapuli, R. (2016). Tools and Techniques for Wt1-Based Lineage Tracing. *Methods in molecular biology (Clifton, N.J.)* **1467**, 41-59.

Second decision letter

MS ID#: dev.204332R1

MS TITLE: Dynamic WT1 expression during gastrulation specifies peritoneal smooth muscle fate independently from mesothelial fate

AUTHORS: Suad Hassan Alsukari, Huei Teng Ng, Lilly Lang, Sharna Lunn, Shanthi Beglinger, Lauren Carr, Michael Boyes, David Andrew Turner and Bettina Wilm

Dear Dr Wilm,

I have now received all the referees reports on the above manuscript, and have reached a decision. The referees' comments are appended below, or you can access them online: please go to .

The overall evaluation is positive and we would like to publish a revised manuscript in *Development*, provided that the referees' very minor comments can be satisfactorily addressed. Please attend to all of the reviewers' comments in your revised manuscript and detail them in your point-by-point response. If you do not agree with any of their criticisms or suggestions explain clearly why this is so. Your manuscript will not require any further review rather I will look it over myself prior to accepting it.

Reviewer 1

Comments for the author

I thank the authors for thoroughly addressing my comments. The revised manuscript is improved in both clarity and scientific depth. I particularly appreciate the addition of the Wt1 ablation data, the reanalysis of some scRNA-seq data, and how the revisions strengthen some of the conclusions. Also, the visual representation in the figures improved, including the consistent inclusion of tamoxifen treatment timelines, and the illustrations in Fig. 2 and 7.

The revised Fig. 3 and corresponding supplementary figures are substantially improved and now clearly highlight the key messages from the public scRNA-seq dataset. I agree that the improved bar plots visualizing the Wt1-expressing cells across embryonic stages is sufficient and indeed the UMAP projections included in the Revision response have not enough resolution. The new of smooth muscle (SMC) and mesothelial (MC) gene signatures, as well as the visualization of their enrichment in Wt1-expressing cells across the early developmental stages, provides support for the proposed early bifurcation of SMC vs MC. The correlation analyses between Wt1, Bra and other signature genes further support the key messages.

The updated Sfig. 3.2 and 3.3 provide important insights into the relationship between Wt1 and Hand2, and the IF co-staining for WT1 and HAND2 is a great addition that strengthens the conclusions by showing co-expression also at the protein level. It is intriguing to see 3 populations of cells in the early LPM: WT1⁺/HAND2⁺, WT1⁺/HAND2⁻, and HAND2⁺/WT1⁻ cells.

The Wt1 ablation experiment (new Fig. 6) is a great addition. It convincingly shows that early loss of Wt1 disrupts vascular SMC organization and vascular network development in the intestine, supporting a direct role for Wt1 in early SMC specification.

Lastly, the revised title reflects indeed better the content of the manuscript.

All major and minor concerns have been fully addressed. Below, I list a few remaining minor points to be addressed in a final round of editing:

Minor edits:

1. Fig. 6: Consider adding a tamoxifen treatment timeline for consistency with other figures (e.g., Fig. 1, 2, and 4).
2. Sfig. 6.1: Panel K should be labeled J', since it shows the same image as J but without annotation.
3. Sfig. 6.3: GFP signal in green is difficult to discern; consider showing the GFP channel in grayscale to improve contrast and visibility of GFP+ cells.
4. Sfig. 8: Please clarify in the legend if the embryo shown in B (and throughout the whole figure?) is the same specimen as the embryo shown in Sfig. 6.1 J-M.

Overall, this is now a strong and well-supported manuscript that contributes valuable new insight into the early specification of peritoneal smooth muscle lineages by Wt1. I thank the authors again for their careful revisions.

Reviewer 2

Advance summary and potential significance to field

The manuscript by Alsukari et al. identifies a novel WT1 expression domain during gastrulation in cells that give rise to smooth muscle cell (SMC) precursors. Tamoxifen-induced lineage tracing at E7.5 or E8.5 showed that WT1-expressing cells contribute to vascular and visceral SMCs in the fetal midgut. Single-cell RNA sequencing and confocal microscopy revealed WT1 expression in the epiblast, primitive streak, emerging mesoderm, and, from E7.5 onward, in the lateral plate mesoderm. Notably, the study shows that vascular and visceral SMC fates are specified independently of the visceral mesothelium. Furthermore, tamoxifen administration at E7.5 to a conditional WT1 knockout mouse model impaired intestinal vascularization by E12.5. Overall, this work offers new insights into the early developmental origins of WT1-specified smooth muscle lineages.

Comments for the author

The authors have addressed most of my concerns and made a significant effort to investigate the putative role of WT1 in early precursors—a point also raised by another reviewer—by generating an early knockout model. Although tamoxifen was administered in the conditional Wt1KO mouse model at E7.5, deletion of WT1 is not immediate. Therefore, I recommend that the authors provide experimental validation confirming WT1 deletion during the gastrulation stage, which is not currently included in the manuscript, or revise the relevant statements accordingly (lines 330-331).

Reviewer 3

Advance summary and potential significance to field

Comments for the author

The authors have adequately addressed the reviewer's comments

Second revision

Author response to reviewers' comments

Comments from the Reviewers:

Reviewer 1: I thank the authors for thoroughly addressing my comments. The revised manuscript is improved in both clarity and scientific depth. I particularly appreciate the addition of the Wt1 ablation data, the reanalysis of some scRNA-seq data, and how the revisions strengthen some of the conclusions. Also, the visual representation in the figures improved, including the consistent inclusion of tamoxifen treatment timelines, and the illustrations in Fig. 2 and 7.

The revised Fig. 3 and corresponding supplementary figures are substantially improved and now clearly highlight the key messages from the public scRNA-seq dataset. I agree that the improved bar plots visualizing the Wt1-expressing cells across embryonic stages is sufficient and indeed the UMAP projections included in the Revision response have not enough resolution. The new of smooth muscle (SMC) and mesothelial (MC) gene signatures, as well as the visualization of their enrichment in Wt1-expressing cells across the early developmental stages, provides support for the proposed early bifurcation of SMC vs MC. The correlation analyses between Wt1, Bra and other signature genes further support the key messages.

The updated Sfig. 3.2 and 3.3 provide important insights into the relationship between Wt1 and Hand2, and the IF co-staining for WT1 and HAND2 is a great addition that strengthens the conclusions by showing co-expression also at the protein level. It is intriguing to see 3 populations of cells in the early LPM: WT1⁺/HAND2⁺, WT1⁺/HAND2⁻, and HAND2⁺/WT1⁻ cells.

The Wt1 ablation experiment (new Fig. 6) is a great addition. It convincingly shows that early loss of Wt1 disrupts vascular SMC organization and vascular network development in the intestine, supporting a direct role for Wt1 in early SMC specification.

Lastly, the revised title reflects indeed better the content of the manuscript.

All major and minor concerns have been fully addressed. Below, I list a few remaining minor points to be addressed in a final round of editing:

Minor edits:

1. Fig. 6: Consider adding a tamoxifen treatment timeline for consistency with other figures (e.g., Fig. 1, 2, and 4).

We have addressed this by adding the schematic as requested by the reviewer.

2. Sfig. 6.1: Panel K should be labeled J', since it shows the same image as J but without annotation.

We thank the reviewer for this suggestion but would like to point out that panels J and K show different embryos in Figure SFig6.1 - this is pointed out in the legend. We therefore would like to leave the panel labelling as is.

3. Sfig. 6.3: GFP signal in green is difficult to discern; consider showing the GFP channel in grayscale to improve contrast and visibility of GFP⁺ cells.

Thanks for pointing this out, we have addressed this and changed the fluorescence channel to grey-scale.

4. Sfig. 8: Please clarify in the legend if the embryo shown in B (and throughout the whole figure?) is the same specimen as the embryo shown in Sfig. 6.1 J-M.

We have addressed this point now in the figure legend of SFigure 8.

Overall, this is now a strong and well-supported manuscript that contributes valuable new insight into the early specification of peritoneal smooth muscle lineages by *Wt1*. I thank the authors again for their careful revisions.

Reviewer 2: SUMMARY OF THE ADVANCE MADE IN THIS PAPER AND ITS POTENTIAL SIGNIFICANCE TO THE FIELD

The manuscript by Alsukari et al. identifies a novel WT1 expression domain during gastrulation in cells that give rise to smooth muscle cell (SMC) precursors. Tamoxifen-induced lineage tracing at E7.5 or E8.5 showed that WT1-expressing cells contribute to vascular and visceral SMCs in the fetal midgut. Single-cell RNA sequencing and confocal microscopy revealed WT1 expression in the epiblast, primitive streak, emerging mesoderm, and, from E7.5 onward, in the lateral plate mesoderm. Notably, the study shows that vascular and visceral SMC fates are specified independently of the visceral mesothelium. Furthermore, tamoxifen administration at E7.5 to a conditional WT1 knockout mouse model impaired intestinal vascularization by E12.5. Overall, this work offers new insights into the early developmental origins of WT1-specified smooth muscle lineages.

SUGGESTIONS TO AUTHORS

The authors have addressed most of my concerns and made a significant effort to investigate the putative role of WT1 in early precursors—a point also raised by another reviewer—by generating an early knockout model. Although tamoxifen was administered in the conditional *Wt1*KO mouse model at E7.5, deletion of WT1 is not immediate. Therefore, I recommend that the authors provide experimental validation confirming WT1 deletion during the gastrulation stage, which is not currently included in the manuscript, or revise the relevant statements accordingly (lines 330-331).

We thank the reviewer for their scrutiny of our interpretation based on the data we have provided. We have now included two additional images to Figure 6 to support the observation that tamoxifen dosing led to Cre activity in the relevant KO embryos. In these embryos (Figure 6B), GFP-expressing cells are shown over the heart (epicardium) and liver/lungs. We acknowledge that this doesn't preclude the possibility that WT1 was not ablated or inactivated in all cells expressing *Wt1* at E7.5 in these embryos. However, as we have pointed out in the discussion, the *Wt1^{CreERT2}* system fails to show 100% efficiency in recombination and it may be the case that there is some variability in loss of WT1 expression within the tissues. We have carefully re-worded this section of the results in the manuscript, lines 330-351, and included some clarifications in the discussion, lines 428-430.

By going over our data again for this revision, we realised that a few minor mistakes had crept in on the reporting on the ablation experiment, involving litter numbers and genotypes of embryos. We have corrected this in the revised version in the results (lines 333-334) and methods sections (line 545).

Reviewer 3: SUMMARY OF THE ADVANCE MADE IN THIS PAPER AND ITS POTENTIAL SIGNIFICANCE TO THE FIELD

SUGGESTIONS TO AUTHORS

The authors have adequately addressed the reviewer's comments

Third decision letter

MS ID#: dev.204332R2

MS TITLE: Dynamic WT1 expression during gastrulation specifies peritoneal smooth muscle fate independently from mesothelial fate

AUTHORS: Suad Hassan Alsukari, Huei Teng Ng, Lilly Lang, Sharna Lunn, Shanthi Beglinger, Lauren Carr, Michael Boyes, David Andrew Turner and Bettina Wilm

Dear Dr Wilm,

I am happy to tell you that your manuscript has been accepted for publication in Development, pending our standard publication integrity checks.